# Preference Learning for AI Alignment: a Causal Perspective

**Katarzyna Kobalczyk** [1]   **Mihaela van der Schaar** [1]

## Abstract

Reward modelling from preference data is a crucial step in aligning large language models (LLMs) with human values, requiring robust generalisation to novel prompt-response pairs. In this work, we propose to frame this problem in a causal paradigm, providing the rich toolbox of causality to identify the persistent challenges, such as causal misidentification, preference heterogeneity, and confounding due to user-specific factors. Inheriting from the literature of causal inference, we identify key assumptions necessary for reliable generalisation and contrast them with common data collection practices. We illustrate failure modes of naive reward models and demonstrate how causally-inspired approaches can improve model robustness. Finally, we outline desiderata for future research and practices, advocating targeted interventions to address inherent limitations of observational data.

## 1. Introduction

The remarkable success of LLMs lies partially in their ability to align with human values, producing responses that are helpful, harmless, and honest. A central method for achieving this alignment is reinforcement learning from human feedback (RLHF), with the LLM's behaviour shaped by reward models derived from datasets of human preferences (Ouyang et al., 2022; Bai et al., 2022). The reward modelling, aka preference learning stage seeks to address a seemingly straightforward question: Given two responses to the same prompt, which one aligns better with human objectives? However, the challenges of this stage are often underestimated, assuming that simple regression-based models fitted to observational datasets can generalise effectively to unseen texts. Meanwhile, evidence suggests this optimism may be misplaced (Tien et al., 2023; Skalse et al.,

2022). Sole reliance on statistical associations observed in the training data is prone to learning rewards that pick up on spurious correlations rather than the true factors influencing user preferences (Singhal et al., 2024; Chen et al., 2024). Latent features of texts often exhibit strong correlations, making it difficult to generalise to examples where such correlations no longer hold. Moreover, pairwise preference datasets are often collected opportunistically, with LLM users both evaluating model responses and generating the prompts eliciting them. As a result, recorded preferences are shaped by the interplay of the LLM's sampling distribution, latent features of the generated responses, and user-specific contexts that vary across the population. These factors raise fundamental concerns about the robustness and reliability of current approaches.

In this work, we argue that building robust reward models requires addressing *what if* type of questions: What would happen if we intervened on specific response characteristics, such as conciseness or creativity? How might preferences change if a different objective were pursued? Current methods are poorly equipped to answer such questions, motivating the need to reframe preference learning through a *causal* perspective. A causal framework enables the disentangling of the effects of different causes on outcomes, facilitating robust predictions under interventions and distribution shifts. Adopting this lens not only offers new insights but also raises pivotal questions: What assumptions are required to generalise across diverse prompts and user groups? How do these assumptions influence data collection practices? How can we address violations of causal assumptions, such as confounding due to user-specific objectives?

**Contributions.** We introduce a causal framework of preference learning for AI alignment, defining the key challenges and identifying critical assumptions for generalising reward models to unseen texts and contexts. Through examples and real-world experiments, we highlight failure modes of naïve models when causal assumptions are violated, including confounding due to user-specific objectives–an issue we are first to identify and address explicitly. We demonstrate how causal representation learning approaches can improve model robustness and propose desiderata for future preference data collection, advocating for targeted interventions to mitigate the inherent limitations of observational data.

[1]Department of Applied Mathematics and Theoretical Physics, University of Cambridge, United Kingdom. Correspondence to: Katarzyna Kobalczyk <knk25@cam.ac.uk>.

*Proceedings of the 42nd International Conference on Machine Learning*, Vancouver, Canada. PMLR 267, 2025. Copyright 2025 by the author(s).

## 2. Background

**Notation.** Throughout this paper, we refer to a random variable with a capital letter (e.g., $X$) and the value it takes as a lowercase letter (e.g., $X = x$). We let $\Sigma^*$ denote the space of natural language. We use coloured boxes to highlight: insights (blue), key assumptions (yellow), theoretical implications (red), and empirical case studies (green).

**Setup.** We consider the standard setup of preference learning for LLM alignment in which we have access to a dataset $\mathcal{D}$ consisting of i.i.d. realisations $(x, y, y', \ell)$ of random variables $(X, Y, Y', L)$, where $X \in \mathcal{X} \subset \Sigma^*$ is the prompt, $Y, Y' \in \mathcal{Y} \subset \Sigma^*$, two candidate responses, and $L \in \{0, 1\}$ is a binary preference label with $L = 1$ indicating that $(X, Y)$ is preferred over $(X, Y')$, denoted by $(X, Y) \succ (X, Y')$, and $L = 0$ the opposite. Here, $\mathcal{X} \times \mathcal{Y}$ denotes the space of all plausible prompt-response of pairs.

**Reward modelling for RLHF.** It is standard to assume that the pairwise preference labels $L$, are dependent on unobservable rewards $R$ and $R'$ assigned internally by the individual to each of $(X, Y)$ and $(X, Y')$, respectively. Rewards are determined by a function $r : \Sigma^* \to \mathbb{R}$, so that $R = r(X, Y)$ and $R' = r(X, Y')$. The likelihood of the option $(X, Y)$ being preferred over $(X, Y')$ is described by the Bradley-Terry-Luce (BTL) model (Bradley & Terry, 1952):

$$P((X, Y) \succ (X, Y')) = \sigma(r(X, Y) - r(X, Y')), \quad (1)$$

where $\sigma$ stands for the sigmoid function. The reward function $r$ is approximated by a parametric model $r_\theta : \Sigma^* \to \mathbb{R}$ whose parameters $\theta$ are chosen by minimising the negative log-likelihood of the observed examples $(x, y, y', \ell) \in \mathcal{D}$:

$$\mathcal{L}(\theta) = - \sum_{(x, y^w, y^\ell) \in \mathcal{D}} \log \sigma(r_\theta(x, y^w) - r_\theta(x, y^\ell)), \quad (2)$$

where $y^w$ indicates the winning response and $y^\ell$ the loosing one. The fitted reward function is subsequently used in the RL stage to provide feedback on generations of the LLM.

**Causality and Potential Outcomes.** The potential outcomes framework (Rosenbaum & Rubin, 1983; Splawa-Neyman et al., 1990) provides a formal approach to causal inference by conceptualising causation in terms of interventions. At its core, the framework models how an outcome of interest would differ under different interventions, enabling reasoning about causal effects. It considers a set of **units** (e.g., individuals) to which a **treatment** or intervention is applied. For each unit, the treatment $T$ can take on different values $t$, where most commonly we have either $T = 1$ for receiving the treatment and $T = 0$ for being in the control group; see e.g., (Lopez & Gutman, 2017) for extensions to multiple treatments. In the case of binary treatments, each unit has two **potential outcomes** denoted as $Y(T = 1)$, or in short $Y(1)$–the outcome if the unit receives the treatment, and $Y(T = 0) \equiv Y(0)$–the outcome if the unit does

not. However, only one of these outcomes is observed (the factual outcome $Y$), while the other (the counterfactual outcome) remains unobserved. The goal of causal inference is to estimate the potential outcomes under both treatments, or their difference $Y(1) - Y(0)$, known as the causal effect.

## 3. The Causal Framework

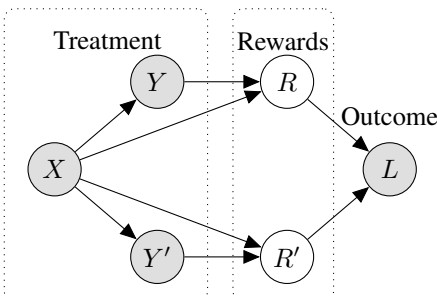

Figure 1: *The causal model of preferences.* Given prompt $X$ and the two responses $Y$, $Y'$ users assigns them unobservable rewards $R$, $R'$ determining the preference label $L$.

We can think of the observed tuples $(X, Y, Y')$ as treatments assigned to human labellers tasked with selecting a response that they prefer and the observed labels $L$ as outcomes. Treatments are assigned according to some (often unknown) propensities: $\pi(x, y, y;) := P(X = x, Y = y, Y' = y')$, where in most cases we have that $Y$ and $Y'$ are conditionally independent given $X$. The distributions $P(Y|X = x)$ and $P(Y'|X = x)$ are not necessarily the same. For instance, given $X$, $Y$ can be sampled from a base LLM policy and $Y'$ from a policy controlled by the researcher in a pre-defined way. We assume that conceptually, each individual providing their preferences is a priori associated with a set of potential outcomes $L(X = x; Y = y, Y = y') \equiv L(x; y, y')$, for any two responses $y, y' \in \mathcal{Y}$, and prompt $x \in \mathcal{X}$. Potential outcomes capture the hypothetical preferences for all pairs of texts that could have been observed. If we had a way of knowing $L(x; y, y')$, for any $x \in \mathcal{X}$ and $y, y' \in \mathcal{Y}$, we could answer counterfactual questions regarding user preferences for hypothetical LLM's responses. However, in reality, we only observe the preference choice associated with the treatments actually received: $(X, Y, Y')$.

We can adopt the potential outcomes framework to the BTL model by introducing the notion of potential rewards, $R(x, y) \equiv R(X = x, Y = y)$ and $R'(x, y') \equiv R'(X = x, Y' = y')$, representing the hypothetical rewards assigned by an individual to any prompt-response pair for $x \in \mathcal{X}$, $y, y' \in \mathcal{Y}$ in our corpus so that:

$$\mathbb{E}[L(x; y, y')] = P(L(x; y, y') = 1)$$
$$= \sigma(R(x, y) - R'(x, y')), \quad (3)$$

Thus, the BTL model directly relates the difference of po-

tential rewards with the expected potential outcome.[1]

> **The fundamental challenge.** The fact that for a given unit we can observe the outcome $L$ only for the observed treatment condition[a] $(X, Y, Y')$ is known as the fundamental challenge of causal inference. Potential outcome prediction is related to the **generalisation** challenge in conventional machine learning terminology—our goal is to predict the effect of hypothetical interventions on LLM's responses, by estimating $\mathbb{E}[L(x; y, y')]$ for any $x \in \mathcal{X}$ and $y, y' \in \mathcal{Y}$.
>
> ---
> [a]or at most a finite set of treatments $\{(X_i, Y_i, Y_i')\}_{i=1}^n$.

### 3.1. What makes potential outcomes identifiable?

Causal inference provides a framework to answer counterfactual, 'what if' questions even when only observational data is available. The key part lies in ensuring that the causal quantity of interest can be estimated from observational data alone. The following commonly made assumptions, adapted to our preference learning setup, are sufficient to guarantee identifiability and non-parametric estimability:

> **Assumption 1** (Consistency). For an individual with prompt-response assignment $(X, Y, Y')$, we observe the associated potential outcome, i.e. $L = L(X; Y, Y')$.

> **Assumption 2** (Unconfoundedness). There are no unobserved confounders, so that $L(x; y, y') \perp\!\!\!\perp (X, Y, Y')$, for all $x \in \mathcal{X}, y, y' \in \mathcal{Y}$.

> **Assumption 3** (Unconditional Positivity). Treatment assignment is non-deterministic, i.e. $0 < P(X = x, Y = y, Y' = y') < 1$ for all $x \in \mathcal{X}$ and $y, y' \in \mathcal{Y}$.

> **Proposition 1.** *Under assumptions (1), (2) and (3), for all $x \in \mathcal{X}$, $y, y' \in \mathcal{Y}$,*
>
> $$\mathbb{E}[L(x; y, y')] = \mathbb{E}[L|X = x, Y = y, Y' = y'],$$
>
> *so that observed statistical associations have a causal interpretation.*
>
> *Proof.* Appendix B. □

---

[1]The notion of potential rewards $R(x, y)$ and $R'(x, y')$ should not be confused with the reward function. $R$ and $R'$ represent two random variables which, in principle, can have distinct distributions so that $P(R(X = x, Y = y)) \neq P(R'(X = x, Y' = y))$– this can be the case, for instance, if the response seen on the left is systematically valued higher than the response seen on the right. Most commonly, it is however assumed that $R = r(X, Y)$ and $R' = r(X, Y')$ for a deterministic reward function $r : \Sigma^* \to \mathbb{R}$ in which case $P(R(X = x, Y = y)) = P(R'(X = x, Y' = y))$.

We note that the absolute values of the potential rewards are, in general, not identifiable (any shift of the reward function by an arbitrary function of the prompt results in the same likelihood). However, we have: $R(x, y) - R(x, y') = \sigma^{-1}\{\mathbb{E}[L(x; y, y')]\} = \sigma^{-1}\{\mathbb{E}[L|X = x, Y = y, Y' = y']\}$, so that the difference in potential rewards is a function of the observable distribution $P(L|X, Y, Y')$, and therefore it is identifiable.

> **Corollary 1.** *The difference in potential rewards $R(x, y) - R'(x, y')$ is identifiable.*

The importance of assumptions 1-3 in enabling causal inferences from observational data makes us questions their feasibility in the context of typical preference data collection. In particular, we identify several potential challenges:

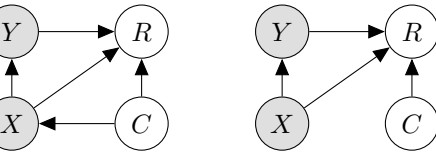

(a) Self-written prompts     (b) Randomised prompts

Figure 2: *Confounding due to user-specific objectives.* a) The user-specific contextual variable $C$ can act as a confounder. If the prompts $X$ are written by the users themselves, $C$ affects the assigned rewards $R, R'$ and partially determines the treatment $(X, Y, Y')$. b) Even if $C$ is not confounding, it may influence the user specific rewards, introducing individual-level variation in treatment effects.

> ♀ **Confounding bias.** The assumption of unconfoundedness is about who scores the texts and not their contents. We stress that this assumption can be easily violated. This may be the case if the prompts $X$ are not randomly assigned, but written by the individuals who then score the generated responses. As a result, users scoring the LLM's responses of one kind may systematically differ from users scoring responses of another kind. We can formalise this by introducing a user-specific contextual variable $C$ that influences both $X$ and $R$ (see Figure 2).

**Example 1** (Violation of Unconfoundedness). Suppose the population consists of medical experts ($C = $ expert) and non-trained users ($C = \neg$expert). Experts are more likely to ask questions about niche topics, leading to complex LLM responses filled with medical jargon, and are also more inclined to prefer such detailed responses due to their professional focus. In this case, the assumption of unconfoundedness would not hold, as the unobservable user-specific variable $C$, indicating the expertise of a user influences both the treatment assignment and their preference choices.

In the face of confounding due to user-specific covariates $C$, it is necessary to adjust for these variables to avoid the bias. Instead of focusing on the average $\mathbb{E}[L(x,y,y')]$, we shall instead consider: $\mathbb{E}[L(x,y,y')|C=c]$. Assumptions 2 and 3 must be then modified accordingly as: 2) $L(x;y,y') \perp\!\!\!\perp (X,Y,Y')|C=c$ and 3) $0 < P(X=x, Y=y, Y'=y'|C=x) < 1$ for all $x \in \mathcal{X}$, $y, y' \in \mathcal{Y}$ and $c \in \mathcal{C}$, where $\mathcal{C}$ is the space of all levels of covariates. Throughout the rest of this work we will use the term *positivity* as a shorthand for the unconditional case and the term *conditional positivity*, aka the *overlap*, will be used when user-specific covariates are considered.

To the best of our knowledge, confounding due to user-specific covariates has not been addressed in prior works on preference learning for AI alignment. We find this issue of a significant importance and we will study it in greater depth in sections 3.2.1 and 4.

**Heterogenous treatment effects.** Even under prompt randomisation, the characteristics of a user or other forms of unobservable context (represented by the variable $C$ in Figure 2) may influence the rewards assigned to each response, leading to heterogeneity in preferences among the users. In this case, implicitly marginalizing over $C$ inflates the variance of the predictions and may mask the true causal relationships present among different subpopulations characterised by distinct $C$'s. Siththaranjan et al. (2024) refer to this variable as the "hidden context" and show that averaging over $C$ is equivalent to adopting the so called *Borda count* rule, which may lead to counter-intuitive results. From the point of view of causal inference, $C$ introduces individual-level variation, leading to heterogenous treatment effects where any given treatment (here, any given response) might affect different users in different ways. If not adequately addressed, this in turn may lead to underrepresentation issues (Wu et al., 2023), resulting in allocation of suboptimal treatments for underrepresented populations. In our context, an LLM finetuned to the average preferences may generate suboptimal responses for certain subgroups of users. This underscores the importance of not only focusing on the population average effects, $\mathbb{E}[L(x,y,y')]$, but also subgroup-level effects: $\mathbb{E}[L(x,y,y')|C=c]$.

The assumption of overlap requires that every user has a non-zero probability of being assigned any combination of texts given their covariates. In conventional, non-parametric treatment effect estimation, this ensures that we have enough data to estimate the causal effects across all levels of treatments and covariates. In our case, however, treatments belong to the extremely high-dimensional space of natural language. We only observe a small subset of potential prompt-response pairs, each often being scored by at most one user, while in general, our goal is to *generalise* to texts not part of the training corpus. With no sufficient overlap

in the observational space generalisation of reward models is possible by assuming that the observed texts can be compressed into a lower-dimensional latent representations capturing all **latent features** of texts that influence the rewards assigned.

### 3.2. Robust Generalisation via Latent Treatments

We present a model of latent treatments allowing us to consider assigned texts not on the literal level, but in terms of their underlying features. This formulation is key to facilitating generalisation to unseen texts and enables the learning of human-interpretable reward models, supporting the design of targeted interventions (see Appendix C). While appealing, the introduction of the latent treatment model presents its own set of challenges. For brevity and clarity, we focus on a single prompt-response pair $(X,Y)$, assuming the same applies to the alternative $(X,Y')$.

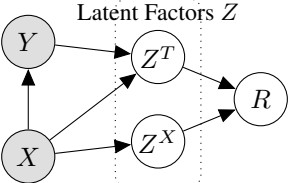

Figure 3: *The latent treatment model.* The effect of observed texts on $R$ can be compressed into a set of latent variables $Z$ partitioned into two kinds: $Z^X$–the artifacts of $X$ and $Z^T$–the latent treatments determined jointly by $X$ and $Y$.

We assume that features of texts that affect the rewards can be effectively summarised into a set of latent features $Z = \{Z_1, \ldots, Z_n\} \in \mathcal{Z}$ split into two parts: $Z^X$–artefacts of the prompt influenced by $X$ only, and $Z^T$–the latent treatments influenced by $(X,Y)$ jointly. $Z^X$ may describe features like the topic of the conversation or the type of task involved. On the other hand, $Z^T$ contains the features of the response $Y$ that cannot be determined without putting them in the context of $X$. This involves, for instance, factual correctness or instruction-following. In our definition, $Z^T$ also subsumes the artifacts of the response that can be extracted from $Y$ without having access to $X$, like its style or length.

Formally, we assume existence of a feature extracting function $g : \Sigma^* \to \mathcal{Z}$, $(X,Y) \mapsto Z = [Z^X, Z^T]$, that can be decomposed into two maps: $g^X : \Sigma^* \to \mathcal{Z}^X$ and $g^T : \Sigma^* \to \mathcal{Z}^T$, s.t. $Z^X = g^X(X)$ and $Z^T = g^T(X,Y)$, $Z'^T = g^T(X,Y')$. The assumption of sufficiency of the latent factors in determining the rewards assigned can be then defined as:

**Assumption 4** (Latent Sufficiency)**.** Assume there exists functions $g^X : \Sigma^* \to \mathcal{Z}^X$ and $g^T : \Sigma^* \to \mathcal{Z}^T$ such that

for any $x \in \mathcal{X}$, $y, y' \in \mathcal{Y}$ we have

$$R(X = x, Y = y) = R(Z^X = z^X, Z^T = z^T)$$
$$R'(X = x, Y = y') = R'(Z^X = z^X, Z'_T = z'_T),$$

where $z^X = g^X(x)$, $z^T = g^T(x, y)$, $z'^T = g^T(x, y')$.

💡 **Why $Z^X$ matters?** If the effect of $Z^X$ on $R$ is *additive* so that for any $x, y$ we have $R(x, y) = f_1(z^T) + f_2(z^X)$, then we could omit $Z^X$ from the model, as it would have no influence on the outcome $L$ defined by $R(x, y) - R'(x, y') = f_1(z^T) - f_1(z'^T)$, $\forall x, y, y'$. Our claim, however, is that the effect of $Z^X$ is rarely of an additive nature and thus, it cannot be disregarded.

**Example 2** (Non-additive effects of $Z^X$). Consider $Z^X$, representing the task type (e.g., summarisation or creative writing), and $Z^T \in \mathbb{R}$, the conciseness of a response. While in general, responses should not be excessively short or excessively long, the optimal level of conciseness varies between the two task types, introducing a non-additive interaction effect between $Z^X$ and $Z^T$ which could be, for instance, described as:

$$R(x, y) = \begin{cases} \beta_0 (z^T - \gamma_0)^2 & \text{if } z^X = \text{summarisation} \\ \beta_1 (z^T - \gamma_1)^2 & \text{if } z^X = \text{creative writing}, \end{cases}$$

where we would expect that $\gamma_0 > \gamma_1$, reflecting that the optimal conciseness level for summarisation is greater (i.e., more concise) than for creative writing.

The information-compressing nature of $g$ makes it possible to estimate the causal effects $\mathbb{E}[L(x; y, y')]$ for prompt-response pairs not previously observed in our training corpus. In particular, instead of requiring that the positivity assumption holds in the observable space, it is sufficient to consider the weaker assumption of **latent positivity**:

**Assumption 5** (Unconditional Latent Positivity). For all $z^X \in \mathcal{Z}^X$ and $z^T, z'^T \in \mathcal{Z}^T$

$$0 < P(Z^T = z^T, Z'_T = z'^T, Z^X = z^X) < 1.$$

**Proposition 2.** *Under assumptions 1, 2, 4, and 5*

$$\mathbb{E}[L(x; y, y')] = \mathbb{E}\left[L | Z^X = z^X, Z^T = z^T, Z'^T = z'^T\right],$$

*for $z^X = g^X(x)$, $z^T = g^T(x, y)$ and $z'^T = g^T(x, y')$.*

*Proof.* Appendix B. □

💡 **The significance of sufficiency and latent positivity** is that, if these assumptions hold, it is possible to estimate the potential outcome for a previously unobserved examples $(x, y, y')$, by only considering the average outcomes of texts observed within our corpus that have the same latent structure as $(x, y, y')$.

### 3.2.1. IMPLICATIONS FOR PRACTITIONERS

💡 **Limited latent positivity.** The assumption of latent positivity requires that all combinations of latent factors need to be observable. However, it may be that certain factors are perfectly correlated with each other, in which case disentangling their effects is not possible. When the positivity is limited, i.e. when strong correlations exists, it hinders the efficiency of estimators (Hahn, 1998; Hirano et al., 2003; Crump et al., 2006), requiring large amounts of data for robust results.

**Example 3** (Correlated latent factors).

*a) Topic and Tone:* Suppose $Z^X$ stands for the topic of the prompt and one of $Z_i^T$'s, indicates the presence of a formal tone. Professional, work-related topics may naturally lead to LLM responses with a formal tone so that $P(Z_i^T = \text{informal}, Z^X = \text{professional})$ and $P(Z_i^T = \text{formal}, Z^X = \text{casual})$ being near 0.

*b) Completeness and Conciseness:* Suppose $Z_1^T$ represents whether a response is complete and $Z_2^T$ indicates whether it is succinct. Complete responses often require elaboration, making succinct yet complete responses rare, i.e., $P(Z_1^T = 1, Z_2^T = 1)$ close to 0.

**Interpretable causal effects.** If the latent factors $Z$ carry a human-interpretable meaning, we can draw further insights regarding the *causal effects* of each individual factor on user preferences. Appendix C.1 defines the appropriate measurement tools. We also make connections on how such causally-interpretable models can enable targeted modifications to LLM's responses and thus enable collection of interventional data (Appendix C.2).

**Discovery of $Z$.** Thus far, we have assumed that the mapping from raw observations to latent causal factors is given. In practice, it needs to be learned from the available data. Since it is not plausible to obtain an exhaustive list of all causal factors and label each example in the corpus with respect to these factors, we must instead shift towards *causal discovery*–which is significantly more challenging (Schölkopf et al., 2021). Discovering $Z$ in a an unsupervised or semi-supervised fashion (i.e., without explicit labels for the latent factors or only with partial labels) creates the risk of *causal misidentification* (Locatello et al., 2019; Makar et al., 2022; Brehmer et al., 2022; Ahuja et al., 2023), where

Table 1: Test time accuracy [%] of the BTL models with varying values of latent positivity in the observational dataset.

| $\rho^{tr}$ | 0.0 | 0.3 | 0.6 | 0.9 |
|---|---|---|---|---|
| ID | $69.3 \pm 0.2$ | $68.3 \pm 0.2$ | $67.8 \pm 0.2$ | $67.6 \pm 0.2$ |
| OOD | $64.5 \pm 0.2$ | $62.0 \pm 0.2$ | $59.7 \pm 0.2$ | $57.8 \pm 0.1$ |

the learned rewards mistakenly rely on spurious features (e.g., length (Singhal et al., 2024; Chen et al., 2024) or formatting (Zhang et al., 2024)) instead of the true causal factors. This in turn leads to robustness issues. The learned latent representations cannot capture features that are spuriously correlated with true causal factors. At the same time, none of the causal factors can be omitted–not measuring all causal factors influencing the rewards will necessarily lead to omitted-variable bias (Fong & Grimmer, 2023).

**Experiment** (Limited latent positivity). We illustrate the significance of the latent positivity by examining its influence on standard BTL models. Using the UltraFeedback dataset (Cui et al., 2024), we consider the *truthfulness* and *instruction following* factors of each prompt-response pair, denoted $Z_1$ and $Z_2$, respectively, and scored from 0 to 5. We construct five training datasets by varying the correlation coefficient $\rho^{tr}$ between $Z_1$ and $Z_2$. We let the reward function be $r(x, y) = \frac{1}{4}z_1 + \frac{3}{4}z_2$, with the true values of $z_1$ and $z_2$ not available for training. The datasets consists of tuples $(x, y, y', \ell)$ with $\ell$ determined by the function $r$. We assess the robustness of the learned reward models to shifts in the correlation of $Z_1$ and $Z_2$, testing on previously unseen examples either from the same distribution as the training examples (ID: $\rho^{test} = \rho^{tr}$), or not, with $\rho^{test}$ being *negative* (OOD: $\rho^{test} < 0$). Refer to Appendix D.2 for details.

As shown in Table 1, performance on unseen ID examples remains high across all $\rho^{tr}$ values, demonstrating that latent sufficiency enables generalisation to unobserved prompt-response pairs. However, with OOD examples breaking the training-time correlation between $Z_1$ and $Z_2$, performance significantly declines due to reduced latent positivity. The ID-OOD accuracy gap widens as $\rho^{tr}$ increases. Even at a moderate $\rho^{tr} = 0.6$, accuracy drops to 59.7%, compared to 67.8% on unseen ID examples (note, the correlation between truthfulness and instruction-following across the entire UltraFeedback dataset is precisely 0.63). Appendix D.1 provides further analysis and a visualisation explaining this behaviour.

We note that this experimental setup does not strictly violate the positivity assumption, making identifiability possible in the infinite data limit. However, perfect correlation is unrealistic in practice. The focus of this experiment is on the statistical challenges posed by limited latent overlap. The observed performance degradation reflects a statistical issue arising from near-violations of the positivity assumption, rather than a fundamental identifiability failure.

💡 **Confounding & Latent Overlap.** The challenges of reward modelling are exacerbated even further when confounding effects are present, in which case we require that $0 < P(Z^X = z^X, Z^T = z^T, Z'^T = z'^T | C = c) < 1$, for all $z^X \in \mathcal{Z}^X, z^T, z'^T \in \mathcal{Z}^T$ and $c \in \mathcal{C}$. If $C$ represents an objective according to which the LLM's responses are evaluated and this objective $C$ also affects the type of prompts that the user writes, then the latent overlap is likely to be particularly limited.

**Example 4.** Imagine two groups of labellers, each with a distinct objective $C \in \{0, 1\}$. For $C = 0$, a labeller is focused on assessing the *helpfulness* of the model, while for $C = 1$, they are focused on assessing the *harmlessness* of the model. These differing objectives lead the two groups to generate prompts with distinct intents: the helpfulness-focused ($C = 0$) produce assistance-related prompts, while the harmlessness-focused ($C = 1$) generate prompts designed to elicit harmful behaviour. Consequently, the prompt distributions $P(X|C = 0)$ and $P(X|C = 1)$ are distinct, potentially leading to overlap violations in $P(Z|C)$. Despite this, we wish to answer interventional questions regarding the preference choices of the helpfulness-focused users ($C = 0$), if presented examples $(x, y, y')$, with $x \sim P(X|C = 1)$ and vice versa (see Figure 4). With no latent overlap, fundamental assumptions of causal inference deem this is an infeasible task. In section 4 we will study this case from an empirical perspective.

Even when the ground-truth latent overlap holds, learning robust representations $\hat{Z}$ from finite datasets poses significant challenges. Correlations between the observed examples $(X, Y, Y')$ and the objectives $C$ in the training data can lead to $\hat{Z}$ entangling features of the text with user-specific objectives. As a result, $\hat{Z}$ may fail to generalise to examples with $X$'s and $C$'s no longer correlated.

## 4. Case Study: The Challenge of Confounding

Modern reward learning systems frequently rely on preference data collected opportunistically–a user types a query, two candidate responses are generated, and the user is asked to select the response they prefer. In such setups, user-specific objectives can act as **confounders**, influencing both the types of prompts the user formulates and the subsequent

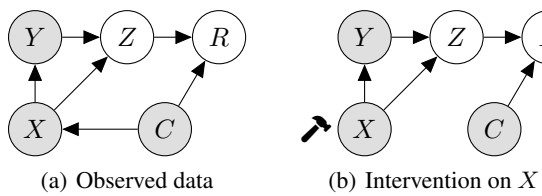

(a) Observed data      (b) Intervention on $X$

Figure 4: A reward model relying on the set of true causal factors $Z$ enables prediction of outcomes under targeted interventions. We wish to predict the rewards when intervening on the prompt distribution $X$, breaking the correlations between prompt type and user-specific objectives $C$.

preference choices. This study highlights the challenges associated with robust reward learning in the presence of such confounding effects. To illustrate these issues, we revisit Example 4 and simulate the described scenario utilising an augmented version of the well-known HH-RLHF dataset, as provided by Siththaranjan et al. (2024).

> The original HH-RLHF dataset (Bai et al., 2022) contains prompt-response pairs categorised into two subsets: *helpful* and *harmless*. Siththaranjan et al. (2024) noted significant distributional differences between the prompts in these two subsets. Meanwhile, all prompt-response pairs in the helpful subset are evaluated based on the objective of helpfulness, while those in the harmless subset are assessed according to the objective of harmlessness. To examine the challenges arising when users operate under potentially conflicting objectives, Siththaranjan et al. (2024) created an augmented version of this dataset, introducing synthetic, counterfactual labels so that texts in the helpful subset are also scored according to the harmlessness objective, and vice versa.
>
> As in Example 4, we let $C \in \{0, 1\}$ encode the objective of the labeller. We also denote by $\text{type}(x) \in \{0, 1\}$, whether a given example $(x, y, y')$ was originally part of the helpful or harmless subset. To control the degree of confounding due to $C$, we define a parameter $\rho$ that measures the alignment between a user's objective and the type of prompts they write: $\rho = P(\text{type}(X) = C)$. At $\rho = 1.0$, the prompt-response distribution recovers the original HH-RLHF dataset, with prompt type and user objective being perfectly correlated. In contrast, at $\rho = 0.5$, the prompt type is fully randomised, eliminating the confounding effect.
>
> **The Goal.** Our aim is to investigate how the confounding effects related to user-specific objective influence reward model performance and to explore mitigation strategies. We seek to answer the interventional questions, such as "What would the preferences of the helpfulness-focused users ($C = 0$) be, have they been presented

examples $(x, y, y')$ with $\text{type}(x) = 1$?". We note that, the two objectives considered are not always aligned–a highly helpful response may not necessarily be harmless–making this a challenging machine learning problem.

**Dataset:** W create six training datasets varying the value of $\rho$. The datasets comprise of tuples $(x, y, y', c, \ell)$, with an equal number of examples for each objective $c \in \{0, 1\}$. Type labels are not part of the training data as they wouldn't be available in a real-world data collection setup. **Models:** We train three multi-objective reward models. The *Base* model corresponds to the most straightforward architecture, where the objective label $c$ is concatenated with the reward model's inputs (Figure 5(a)). In contrast, the *Multihead* model is inspired by the ground-truth causal graph wherein the latent factors $Z$ are conditionally independent of $C$ given $(X, Y)$. The model learns a latent representation of the prompt-response pairs, $\hat{Z}$, passed to two independent prediction heads corresponding to the two objectives (Figure 5(b)). Finally, the proposed *Adversarial* model (Figure 5(c)) incorporates an additional adversarial objective (Ganin et al., 2016) to regularise representation learning. This is inspired by recent advancements in representation learning for treatment effect estimation (Bica et al., 2020; Ozery-Flato et al., 2020; Du et al., 2021) (see section 4.1 for details). **Evaluation:** Trained models are evaluated on unseen prompt-response pairs derived from either of the data subsets and labelled according to objectives both consistent and inconsistent with their prompt type. Refer to Appendix E.1 for more details.

### 4.1. The Adversarial Multi-objective Reward Model.

Due to the training time correlation between $\text{type}(X)$ and $C$ the learned latent representations may mistakenly treat the information about $\text{type}(X)$–easily extractable from the observations $(X, Y)$–as having a causal effect on $R$. The goal of the proposed adversarial training method is to make $\hat{Z}$ not predictive of $\text{type}(X)$, while retaining all features that causally influence $R$. Let $g_\theta$ represent the network learning a latent representation $\hat{Z}$ from $(X, Y)$, s.t. $\hat{z} = g_\theta(x, y)$ for all $x, y \in \mathcal{X} \times \mathcal{Y}$. In the *Multihead* reward model, the rewards are obtained by passing the latent $\hat{z}$ to the respective reward head $f_{w_0} : \hat{\mathcal{Z}} \to \mathbb{R}$ or $f_{w_1} : \hat{\mathcal{Z}} \to \mathbb{R}$ so that:

$$r_{\theta, w_0, w_1}(x, y, c) = \begin{cases} f_{w_0}(g_\theta(x, y)) & \text{if } c = 0 \\ f_{w_1}(g_\theta(x, y)) & \text{if } c = 1 \end{cases} \quad (4)$$

The *Adversarial* model introduces an additional network $h_\phi : \mathcal{Z} \to \{0, 1\}$ whose goal is to predict the objective label $C$ from $Z$ as a proxy for $\text{type}(X)$. Adversarial training enourages $g_\theta$ to discard the spurious information about $\text{type}(X)$ while retaining causal features relevant to the out-

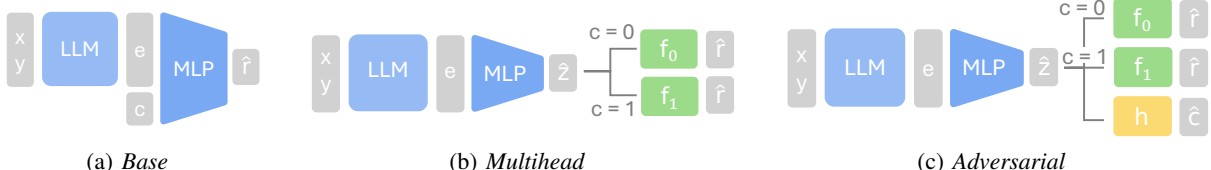

| (a) *Base* | (b) *Multihead* | (c) *Adversarial* |

Figure 5: *Comparison of model architectures.*

come. The training objective for this model is:

$$\min_{\theta, w_0, w_1} \max_{\phi} \mathcal{L}_{\mathrm{R}}(\theta, w_0, w_1) - \lambda \mathcal{L}_{\mathrm{adv}}(\theta, \phi), \qquad (5)$$

where $\mathcal{L}_{\mathrm{R}}(\theta, w_0, w_1)$ is the standard BTL loss for the reward function $r_{\theta, w_0, w_1}$, the second term $\mathcal{L}_{\mathrm{adv}}(\theta, \phi)$ is the binary cross-entropy loss between the true $c$'s and their log-probabilities predicted by $h_\phi \circ g_\theta$, and $\lambda$ is a hyperparameter balancing the two objectives. Appendix E.1 contains further details regarding implementation and training.

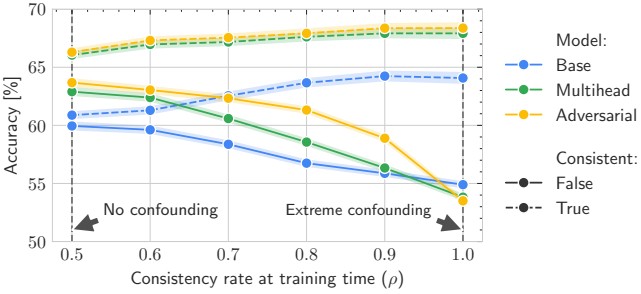

Figure 6: *Test time accuracy vs. confounding*. The label "consistent" indicates whether $\mathrm{type}(X) = C$. The causally-inspired multihead architecture with additional adversarial balancing significantly reduces overfitting and improves generalisation to the inconsistent OOD examples.

### 4.2. Case Study Results

We analyse the results of the experiments summarised in Figure 6 and presented in detail in Appendix E.2.

▶ **No overlap ⇒ failure to generalise.** When the objective is fully determined by the data subset ($\rho = 1.0$), all three models exhibit substantial overfitting to the training distribution. While the performance on the unseen, consistent samples is high–with the multihead and adversarial architectures showing modest improvements–all models fail to generalise to inconsistent examples, with their accuracies falling just below 55%. This outcome is expected, as the latent overlap assumption is likely violated at $\rho = 1.0$. Higher $\rho$ values lead to training sets with fewer inconsistent examples, making models more susceptible to overfitting and causal misidentification. ▶ **Increasing the overlap helps.** As expected,

accuracy on inconsistent test samples improves across all models as $\rho$ decreases in the observational training data. ▶ **Extra gains from causally-inspired models.** We observe that the *Multihead* architecture significantly outperforms the *Base* model, supporting our hypothesis that separation of latent representation learning from objective-conditioned reward prediction enhances robustness to training-time correlations between prompt types and objectives. Introducing the adversarial objective further strengthens this effect, significantly boosting accuracy, especially in strong confounding regimes.

## 5. Conclusions & Limitations

Below, we summarise key conclusions from this work, contextualise them within existing literature, and outline future research directions. Appendix A contains an extended discussion of the related work.

**The importance of data collection.** This study underscores the critical role of data collection mechanisms in reward learning for preference modelling. Controlled, randomised experiments, where prompt-response pairs are allocated randomly across a representative population serve as the gold standard. However, in practice, preference data is often collected opportunistically, relying on user-written prompts and LLM-generated responses. This approach introduces the risk of confounding, where users' latent objectives influence both the queries they pose and the feedback they provide, as demonstrated in our case study.

**The challenge of unobserved confounding.** Our experiments assumed explicit access to user-specific objectives, allowing us to simulate and control for confounding effects. While this setup provided a clear experimental framework, it does not fully reflect real-world scenarios, where user objectives are not directly observable, posing the challenge of unobserved confounding. To address this, preference data collection methods could be enhanced to infer user-specific objectives through, e.g. a) **explicit feedback**: users could provide rationales for their preferences, offering richer insights into their underlying objectives; **b) auxiliary data:** preferences could be inferred from contextual information, such as user demographics or historical interactions. Only a limited number of existing works (Li et al., 2024; Wu et al., 2024; Liu et al., 2024; Kobalczyk et al., 2024) have

Code for reproducing the experiments is made available at: https://github.com/kasia-kobalczyk/causal-preference-learning.

considered incorporating user-specific information for personalised alignment, yet none have explicitly examined the confounding issues identified in this work.

**Mitigating low overlap.** Our findings underscore the importance of latent overlap. Strong correlations between latent response features can cause catastrophic overfitting, hindering generalisation under distribution shifts. However, controlling overlap in practice is not straightforward. Existing work on robust reward modelling mainly addresses biases from response-specific artefacts, such as length (Singhal et al., 2024; Chen et al., 2024) or formatting (Zhang et al., 2024), but as demonstrated, causal misidentification extends beyond these cases. While approaches derived from causal representation learning can help, architectural modifications have inherent limitations. Instead, we advocate for a shift in preference data collection practices—from passive data gathering to targeted interventions wherein latent factors are systematically controlled to reduce model uncertainty about the set of true causal features. Appendix C.2 outlines potential research directions in this pursuit.

**Limitations.** We note that the assumptions discussed in this paper are *sufficient*–but not strictly *necessary*–for identifiability. In practice, identifiability may still be achievable under weaker conditions, particularly when data from multiple environments is available (Ahuja et al., 2023; Richens & Everitt, 2024; Von Kügelgen, 2023). This is especially pertinent to recent work on reward learning across diverse datasets and personalised reward modelling–areas that have garnered increasing attention in alignment research (Wang et al., 2024; Ramé et al., 2023; Bose et al., 2025). A comprehensive treatment of robust reward learning in the multi-dataset setting is left for future work.

## Impact Statement

This paper presents work whose goal is to advance the field of Machine Learning. There are many potential societal consequences of our work, none which we feel must be specifically highlighted here.

## Acknowledgments

This work was supported by Azure sponsorship credits granted by Microsoft's AI for Good Research Lab. Katarzyna's Kobalczyk research is supported by funding from Eedi.

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

# A. Related Work

**Causal Inference for Text Data and LLM Alignment.** Causality has gained increasing attention in NLP, with several studies proposing methods for treatment effect estimation from text data, each with a different focus. Examples include the discovery of latent text attributes (Fong & Grimmer, 2016), the impact of unmeasured latent treatments (Fong & Grimmer, 2023), non-parametric treatment effect estimation (Pryzant et al., 2021), and the robustness of such estimators (Gui & Veitch, 2023). These works predominantly address data analysis tasks where causality is used as an interpretability tool. In large language models (LLMs), Vig et al. (2020) use causal mediation analysis to study gender biases, while Cao et al. (2022) analyse prompt-based probing from a causal perspective. Other works, such as Wang et al. (2023), introduce in-context causal interventions to alleviate entity biases in prompts, and Hu & Li (2021) adopt causality for controllable text generation. However, none of these directly apply to the pairwise preference learning setup explored in our work or the subsequent LLM fine-tuning stage via RLHF. Few notable exceptions addressing our call for adapting a causal perspective for preference learning and alignment involve the work of Xia et al. (2024) who attempt to leverage a pre-trained reward model as an instrumental variable for causal intervention on LLMs. Lin et al. (2024) draw on importance weighting and double robustness principles to present methods for more robust preference optimization in DPO. Reber et al. (2025) introduce a causal framework for understanding and evaluating spurious correlations of reward models. Butcher (2024) use counterfactual pairs to address spurious correlations during the alignment process. Finally, Liu et al. (2025) propose a causal framework for learning preferences independent of response artifacts (e.g., length), assuming that prompt-independent features are spurious, and introduce a data augmentation technique to eliminate them.

**Reward Hacking and Causal Confusion**. Many prior works in reinforcement learning have studied reward hacking (Skalse et al., 2022), closely linked to causal misidentification (de Haan et al., 2019; Tien et al., 2023). This occurs when learned policies or rewards achieve high accuracy within their training distribution but fail to generalise to novel scenarios because the models do not correctly identify the underlying causal structure. In preference learning for AI alignment, this issue has primarily been explored in the context of response-specific biases, such as length formatting (Zhang et al., 2024) or response length itself (Singhal et al., 2024; Chen et al., 2024; Park et al., 2024), where authors demonstrate that learned reward models or downstream policies improve significantly by simply increasing response length, rather than considering other relevant features. Some evidence suggests that reward model ensembles can improve the robustness of LLM policies (Annervaz et al., 2018; Coste et al., 2024; Eisenstein et al., 2024), but they do not fully eliminate the reward hacking problem, leaving it an open challenge.

# B. Identifiability Proofs

For convenience, we restate the assumptions enabling identifiability of $\mathbb{E}[L(x'y, y')]$.

**Assumption 1** (Consistency). For an individual with prompt-response assignment $(X, Y, Y')$, we observe the associated potential outcome, i.e. $L = L(X; Y, Y')$.

**Assumption 2** (Unconfoundedness). There are no unobserved confounders, so that $L(x; y, y') \perp\!\!\!\perp (X, Y, Y')$, for all $x \in \mathcal{X}$, $y, y' \in \mathcal{Y}$.

**Assumption 3** (Unconditional Positivity). Treatment assignment is non-deterministic, i.e. $0 < P(X = x, Y = y, Y' = y') < 1$ for all $x \in \mathcal{X}$ and $y, y' \in \mathcal{Y}$.

**Proposition 1.** *Under assumptions (1), (2) and (3), for all $x \in \mathcal{X}$, $y, y' \in \mathcal{Y}$,*

$$\mathbb{E}\left[L(x; y, y')\right] = \mathbb{E}\left[L | X = x, Y = y, Y' = y'\right],$$

*so that observed statistical associations have a causal interpretation.*

*Proof.*

$$\begin{aligned}
\mathbb{E}\left[L(x; y, y')\right] &= \\
&= \mathbb{E}(L(x; y, y') | X = x, Y = y, Y' = y') \quad \text{(by positivity and unconfoundedness)} \\
&= \mathbb{E}\left[L | X = x, Y = y, Y' = y'\right] \quad \text{(by consistency)}
\end{aligned}$$

$\square$

As outlined in the main body of the paper, the assumption of overlap in the observational space can be replace with the assumptions of latent sufficiency and latent overlap:

**Assumption 4** (Latent Sufficiency). Assume there exists functions $g^X : \Sigma^* \to \mathcal{Z}^X$ and $g^T : \Sigma^* \to \mathcal{Z}^T$ such that for any $x \in \mathcal{X}, y, y' \in \mathcal{Y}$ we have

$$R(X = x, Y = y) = R(Z^X = z^X, Z^T = z^T)$$
$$R'(X = x, Y = y') = R'(Z^X = z^X, Z'_T = z'_T),$$

where $z^X = g^X(x)$, $z^T = g^T(x, y)$, $z'^T = g^T(x, y')$.

**Assumption 5** (Unconditional Latent Positivity). For all $z^X \in \mathcal{Z}^X$ and $z^T, z'^T \in \mathcal{Z}^T$

$$0 < P(Z^T = z^T, Z'_T = z'^T, Z^X = z^X) < 1.$$

**Proposition 2.** *Under assumptions 1, 2, 4, and 5*

$$\mathbb{E}\left[L(x; y, y')\right] = \mathbb{E}\left[L | Z^X = z^X, Z^T = z^T, Z'^T = z'^T\right],$$

*for $z^X = g^X(x)$, $z^T = g^T(x, y)$ and $z'^T = g^T(x, y')$.*

*Proof.* Let $x \in \mathcal{X}, y, y' \in \mathcal{Y}$ and let $z^X = g^X(x)$, $z^T = g^T(x, y)$, $z'^T = g^T(x, y')$. Since $L = f(R - R', U)$, where $U$ is an independent exogenous noise variable[2] and due to the sufficiency condition, we have that:

1) Consistency in the observational space implies consistency in the latent space, i.e. for an individual with prompt-response assignment $(X, Y, Y')$ whose latent factors are $(Z^X, Z^T, Z'^T) \equiv (g^X(X), g^T(X, Y), g^T(X, Y'))$, we observe the associated potential outcome, i.e. $L = L(Z^X, Z^T, Z'^T)$.

2) Unconfoundedness in the observational implies unconfoundedness in the latent space, i.e. $L(Z^X = z^X; Z^T = z^T, Z'^T = z'^T) \equiv L(z^X; z^T, z'^T)$ is independent of $(Z^X, Z^T, Z'^T)$.

3) $\mathbb{E}\left[L(x; y, y')\right] = \mathbb{E}\left[L(z^X; z^T, z'^T)\right]$

Thus, it follows that:

$$
\begin{aligned}
\mathbb{E}\left[L(x; y, y')\right] &= \mathbb{E}\left[L(z^X, z^T, z'^T)\right] && \text{(by sufficiency)} \\
&= \mathbb{E}\left[L(z^X, z^T, z'^T) | Z^X = z^X, Z^T = z^T, Z'^T = z'^T\right] && \text{(by latent overlap \& unconfoundedness)} \\
&= \mathbb{E}\left[L | Z^X = z^X, Z^T = z^T, Z'^T = z'^T\right] && \text{(by latent consistency)}
\end{aligned}
$$

$\square$

## C. Extended Discussion

### C.1. Interpretability of Latent Factors

Aside from estimating the potential outcomes $L(x, y, y')$ or rewards $R(x, y)$ for any $x \in \mathcal{X}$ $y, y' \in \mathcal{Y}$, we may wish to consider the average effects of their underlying latent factors. If the individual components $Z_k$ of the set of latent factors $Z$ enjoy a human-interpretable meaning, we would like to predict how an intervention of one of these components (e.g. increasing creativity of an answer, or its length) will affect user preferences while holding other features constant. If we assume that each of $Z^k$'s can be represented as a binary value indicating the presence or absence of a particular feature, then

---

[2]The re-expression of $L$ as a function of $R - R'$ under the BTL model is possible thanks to the Gumbel-softmax trick (Luce, 1959; Maddison et al., 2014; Oberst & Sontag, 2019).

following the convention adopted by Fong & Grimmer (2016; 2023) such inferences are made possible by considering a version of the Average Marginal Component Effect (AMCE) adopted to our pairwise preference learning setup:

$$\text{AMCE}_k := \int_{z \in \mathcal{Z}^{-k}} \mathbb{E}\big[L(Z^k = 1, Z'^k = 0, Z^{-k} = z, Z'^{-k} = z\big] m(z) dz, \tag{6}$$

where $\mathcal{Z}^{-k}$ denotes the latent space $\mathcal{Z}$ except the $k$-th one and $m(\cdot)$ is some analyst-defined density on this space, which can be taken, e.g. as a uniform distribution or the distribution induced by the observable $(X, Y, Y')$. The sufficiency and latent overlap conditions imply that this measure is identifiable from observational data.

### C.2. A Roadmap for Robust and Interpretable Alignment

Despite the appealing interpretability properties of the AMCE, similarly to direct potential outcome estimation, it assumes that the mapping $g : \Sigma^* \to \mathcal{Z}$ between the observable texts and their latent features is given and that it captures *all* factors that causally influence the reward, while disregarding any spuriously correlated features. In practice, the map $g$ needs to be learned from data and ideally, with minimal supervision–i.e., not requiring humans to label large numbers of prompt-response pairs according to a prohibitively long list of factors that can plausibly be causally linked to the outcomes. To enable learning of more robust and interpretable feature-extracting and reward predicting functions we outline some key directions for future research and data collection practices.

**Collection of rationales, not only preference choices.** Current preference learning frameworks typically collect binary comparisons between responses, but these do not reveal why a particular choice was made. This lack of transparency obscures the true causal mechanisms driving user preferences and makes it difficult to disentangle spurious correlations from genuine reward-relevant features. Instead of solely collecting pairwise preference choices, future data collections practices could incorporate rationale elicitation, with users providing short explanations for their decisions. Such rationales could be used as weak supervision signals, guiding the learning of latent causal representations without requiring an exhaustive, predefined list of relevant features. This additional form of supervision could also for interpretable post-hoc causal analyses via causal estimands like the AMCE.

**Active querying strategies and interventions.** Much theoretical work has been done demonstrating how identification of causal representations requires auxiliary labels (Locatello et al., 2019; Makar et al., 2022; Brehmer et al., 2022; Ahuja et al., 2023). To gather such labels, rather than querying users at random points during the conversation and generating responses with unknown latent features, active querying strategies (Melo et al., 2024; Muldrew et al., 2024) can improve the efficiency and robustness of preference learning by selecting data points that maximize information gain. This applies to both feature identification (learning to predict $\hat{z}$'s) and reward prediction (learning to predict rewards from $\hat{z}$'s). Queries for rationales should prioritize instances where the model is least confident the feature-extracting part, reducing ambiguity of the learned representations. Reducing the ambiguity of the reward prediction part should benefit from interventional data collection so that given a prompt, two responses are generated that differ in isolation by a specific latent factors, allowing for direct causal attributions. Such interventions should also take into the account user-specific contextual variables to ensure that the latent treatments are well-balanced across different demographic subgroups.

**Interpretable control over LLM-generated content.** To enable targeted and interpretable interventions as described above, it is necessary that LLM-generated content can be explicitly controlled (Hu & Li, 2021; Liang et al., 2024; Dekoninck et al., 2024) to specify desired properties of text . Without such form of control, intervention-based preference learning becomes infeasible, as models would lack the ability to systematically vary latent factors. Language models should be designed to generate responses that explicitly vary along key latent dimensions, such as response verbosity or style, rather than letting these factors emerge implicitly.

Preference learning and optimisation remain an exciting field whose success necessarily relies on integrating multiple approaches, with causality playing a central role. By moving beyond passive preference collection to rationale-aware learning, active data querying, and attribute-conditional control, we can train models that are more robust, interpretable, and generalisable to unseen settings. This shift is crucial for aligning AI systems with human values while avoiding the negative effects of causal misidentification or confounding.

## C.3. The connection to conventional treatment effect estimation

Our formalism follows the potential outcomes framework (Rosenbaum & Rubin, 1983; Splawa-Neyman et al., 1990), but applied to preference learning rather than traditional treatment effect estimation.

In classical causal inference with binary treatments (e.g., drug trials), we define a treatment assignment $T \in \{0, 1\}$ and the observed outcomes $Y$, as well as the potential outcomes under treatment ($Y(1)$) and potential outcomes under no treatment ($Y(0)$). We then aim to estimate treatment effects defined as: $\mathbb{E}[Y(1) - Y(0)]$. The fundamental challenge of causality lies in the fact that for a given individual we only observe one outcome, i.e. if their treatment assignment is $T = t$, then we observe $Y = Y(t)$, but not the counterfactual outcome.

Our framework extends this to preference learning by treating each prompt-response pair $(x, y, y')$ as a distinct "treatment". Here, the tuple $(X, Y, Y')$ is the random variable representing the treatment assignment, and $L(x; y, y')$ represents the potential preference outcome when $(X, Y, Y') = (x, y, y')$, i.e. when the user observes $(x, y, y')$. However, just as in traditional causal inference, we only observe the outcomes for the assigned texts and not all possible texts–a given user is only shown one (or at most a finite subset of) possible prompts and responses. The observed label, defined as $L$, satisfies the consistency condition so that if $(X, Y, Y') = (x, y, y')$, then $L = L(x; y, y')$. The key difference from conventional treatment effect estimation is that while drug studies typically focus on binary treatment effects, we operate in an extremely high-dimensional space of natural language where each $(x, y, y')$ combination is effectively a unique treatment. Rather than computing pairwise contrasts between all possible treatments, we focus on the expected potential preference $\mathbb{E}[L(x, y, y')]$ for any given tuple $(x, y, y')$. This is analogous to estimating $\mathbb{E}[Y(t)]$ for each treatment $t$ in a multi-arm trial, which in turn enables making relative comparisons $\mathbb{E}[Y(t_1) - Y(t_2)]$ between different treatment choices $t_1, t_2$. In our framework, for instance, we could compare $\mathbb{E}[L(x, y_0, y_1)]$ against $\mathbb{E}[L(x, y_0, y_2)]$ to study how the expected preference change if the second observed response $Y'$ is set to $y_2$ instead of $y_1$, while keeping the prompt $X = x$ and the first response $Y = y_0$ fixed. Given the vast space of possible relative comparisons we direct our attention to just the expected potential outcomes, $\mathbb{E}[L(x, y, y')]$, rather than their pairwise differences.

# D. The UltraFeedback Case Study

## D.1. Case Analysis

Suppose that the latent factor $Z$ determining the rewards are two dimensional taking values in $\mathbb{R}^2$ (e.g. $Z$ may correspond to the extent of truthfulness and instruction-following of a candidate response). Suppose that the distribution of $X$'s $Y$'s and $Y''$s is such that $Y \perp\!\!\!\perp Y'|X$ and $Y, Y'$ are identically distributed given $X$. Further, assume that $P(X, Y, Y')$ induces a distribution on $P(Z, Z')$ such that $Z, Z' \overset{iid}{\sim} \mathcal{N}(\mu, \Sigma)$. As a result, we have:

$$\boldsymbol{\delta} := [Z - Z'] \sim \mathcal{N}\left(\begin{bmatrix} 0 \\ 0 \end{bmatrix}, \begin{bmatrix} \sigma_1 & \rho \\ \rho & \sigma_2 \end{bmatrix}\right), \tag{7}$$

where WLOG $\sigma_1 = \sigma_2 = 1$.

Now, suppose that the reward function is linear in $Z_1$ and $Z_2$, so that for all $x \in \mathcal{X}, y \in \mathcal{Y}$, we have

$$r(x, y) = r(z) = \alpha z_1 + (1 - \alpha)z_2 \quad \text{for } \alpha \in [0, 1]. \tag{8}$$

Then, the preference label is completely determined by the vector $\boldsymbol{\delta}$. With $\delta_1 := Z_1 - Z_1'$ and $\delta_2 = Z_2 - Z_2'$, we have that for any samples for which $\alpha\delta_1 + (1 - \alpha)\delta_2 > 0$ the first option is preferred, and vice versa. This can be visualised in the $(\delta_1, \delta_2)$-plane, as in Figure 7.

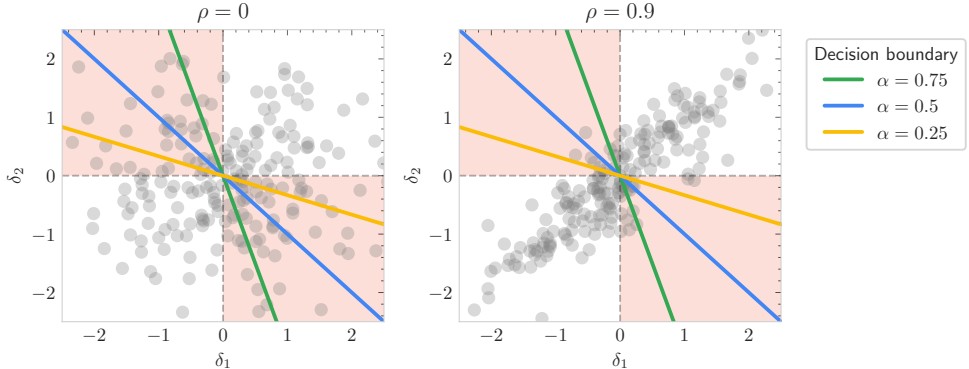

Figure 7: *The impact of $\rho$ on determining $\hat{\alpha}$ and the classification accuracy.*

The decision boundary determining the preference label $L$ is dependent on the ground-truth value of $\alpha$. It is defined by a straight line with a slope of $-\frac{\alpha}{1-\alpha}$ and the intercept at the origin. In a noise-free setting, examples $(x, y, y', \ell)$ for which $\boldsymbol{\delta}$ falls below the decision boundary are labelled with $\ell = 0$ (i.e, the first option $(x, y)$ wins) and all samples above the decision boundary have the label $\ell = 1$ (i.e., the second option $(x, y')$ wins). An optimal fitted reward function $\hat{r}(z) = \hat{\alpha}z_1 + (1 - \hat{\alpha})z_2$ is s.t. $\hat{\alpha} = \alpha$. Samples falling into the first and third quadrants of the $(\delta_1, \delta_2)$-plane are classified correctly *no matter* if the fitted decision boundary (determined by $\hat{\alpha}$) aligns with the ground-truth boundary (determined by $\alpha$). Samples falling into the second and fourth quadrants (red-shaded region) are prone to *misclassification*, if the fitted and ground-truth decision boundaries diverge. If the training data exhibits a high correlation (see the plot with $\rho = 0.9$) few samples fall into the red-shaded region (the probability of this event is precisely $\frac{1}{2} - \frac{\arcsin(\rho)}{\pi}$), making $\hat{\alpha}$ sensitive to outliers and exhibiting a higher variance than for non-correlated samples.

## D.2. Experimental Details

Code for reproducing the experiments is made available at: https://github.com/kasia-kobalczyk/causal-preference-learning.

To emulate the setting described above, we rely on the UltraFeedback dataset (Cui et al., 2024) containing prompt-response pairs scored according to the objectives of honesty, helpfulness, truthfulness and instruction-following, each represented as scalar values between 0 and 5. For simplicity, we focus on just the last two features: truthfulness and instruction-following, as the ground-truth causal factors determining the rewards.

**Datasets.** We create a large dataset of candidate pairs $(x, y, y', z, z', \ell)$, where $z$ and $z'$ are 2-dimensional vectors corresponding to the truthfulness and instruction-following scores of the of the candidate options $(x, y)$ and $(x, y')$, respectively.

The label $\ell$ is defined based on the value of $r(x,y) - r(x,y')$, with

$$r(x,y) = r(z) = \alpha z_1 + (1-\alpha)z_2, \qquad (9)$$

where we set $\alpha = 0.25$. For examples with $r(x,y) - r(x,y') > 0$ we set $\ell = 0$, for examples with $r(x,y) - r(x,y') < 0$ we set $\ell = 1$ and for those with $r(x,y) - r(x,y') = 0$ we sample $\ell$ at random. Based on this large data set containing all prompts and responses in the UltraFeedback dataset we create 4 training subsets, controlling the value of the correlation $\rho^{tr}$ between $z_1 - z'_1$ and $z_2 - z'_2$, where $z_1, z'_1$ represents the truthfulness score and $z_2, z'_2$ the instruction-following score. *Training:* With stratified sampling based on the value of $z - z'$, we create 4 training datasets with 15.000 samples, one for each value of $\rho^{tr} \in \{0.0, 0.3, 0.6, 0.9\}$. The resulting training sets contain tuples $(x,y,y',\ell)$–the ground-truth values of the latent factors are not available during training and need to be approximated in an unsupervised fashion. *Validation:* Validation splits used to determine the optimal stopping point during training of the reward models are samples according to the same distribution as the training datasets and contain 2.000 samples. *Testing:* For each value of $\rho^{tr}$ we also create a testing dataset with matching value of the correlation coefficient (ID), and a testing dataset with only samples falling into the second and fourth quadrants in the $(\delta_1, \delta_2)$-plane, resulting in a correlation coefficient of -0.8 (OOD). The testing datasets are always disjoint from the training examples and contain 15.000 samples.

**Reward model training.** We fit a standard BTL model by passing each prompt-response pair $(x,y)$ through an LLM to obtain its embedding $e$. We do not perform LLM fine-tuning and simply fit the reward models on the pre-computed embeddings. In this experiment we use embeddings of the Llama-3-8B[*] model. The LLM embeddings are processed with a 3-layer MLP with a hidden dimension 512 and an output size of 64. The last linear layer maps from the 64-dimensional latent embedding to a scalar value representing the reward. All models are trained by minimising the empirical estimate of the negative-loglikelihood loss according to the BTL model:

$$\mathcal{L}_R = -\sum_{(x,y^w,y^\ell)} \log \sigma \left( r_\theta(x, y^w) - r_\theta(x, y^\ell) \right), \qquad (10)$$

where $y^w, y^\ell$ stand for the winning and loosing responses, respectively. All models are trained using the Adam optimiser with a learning rate of 1e-4 for 10 epochs. Model weights $\theta$ with the highest validation accuracy are saved for evaluation. For each value of $\rho^{tr}$ we train models with 3 random seeds.

## E. The HH-RLHF Case Study

### E.1. Experimental Details

Code for reproducing the experiments is made available at: https://github.com/kasia-kobalczyk/causal-preference-learning.

**Dataset.** We rely on the extended version of the HH-RLHF dataset (Bai et al., 2022) as provided by Siththaranjan et al. (2024). The entire dataset can be represented as tuples $(t, x, y, y', c, \ell)$, where $c \in \{0, 1\}$ denotes the objective with which the choice $\ell \in \{0, 1\}$ is made and $t \in \{0, 1\}$ denotes the type of $x$, i.e. whether $(x, y, y')$ was originally part of the helpful ($t = 0$) or harmless split ($t = 1$). In the original data (Bai et al., 2022) we only observe examples with $t = c$. Siththaranjan et al. (2024) augment this dataset with counterfactual labels for such that $t \neq c$ which we refer to as inconsistent samples. We create six independent training datasets, controlling the ratio of consistent to inconsistent samples, i.e. the parameter $\rho = P(\text{type}(X) = C) \in \{0.5, 0.6, 0.7, 0.8, 0.9, 1.0\}$. The resulting training datasets consist of 30.000 samples $(x, y, y', c, \ell)$, with the label $t$ not being part of the training sets. We also create validation splits with the same values of $\rho$'s of 6.000 sample. The remaining 46518 samples is left for testing.

**Models.** We train three different versions of multi-objective BTL models. Each model has the same pre-processing backbone. Prompt-response pair $(x, y)$ are passed through an LLM to obtain its embedding $e$. We do not perform LLM fine-tuning and simply fit the reward models on the pre-computed embeddings. We use embeddings of the Llama-3-8B[*] model.

- **Base:** The embeddings $e$ are concatenated with the objective label $c$ and passed through a 3-layer MLP $r_\theta : \mathbb{R}^d \to \mathbb{R}$ with a hidden dimension of 512 and outputting the predicted scalar reward $\hat{r}$. The model is trained by finding parameters $\theta$ that maximise the log-likelihood under the BTL model as in (10).

- **Multihead:** The embeddings $e$ are passed through a 3-layer MLP $g_\theta : \mathbb{R}^d \to \mathbb{R}^{512}$ with a hidden dimension of 512 mapping them to a latent representation $\hat{z} \in \mathbb{R}^{512}$. Depending on the value of $c$, the vectors $\hat{z}$ are then passed to one of

---

[*] huggingface.co/meta-llama/Meta-Llama-3-8B

the two prediction heads $f_{w_0} : \mathbb{R}^{512} \to \mathbb{R}$ or $f_1 : \mathbb{R}^{512} \to \mathbb{R}$. Here, we let $f_0$ and $f_1$ be 1 layer MLPs with a hidden dimension of 512. Thus the reward function can be defined as:

$$
r_{\theta, w_0, w_1}(x, y, c) = \begin{cases} f_{w_0}(g_\theta(x, y)) & \text{if } c = 0 \\ f_{w_1}(g_\theta(x, y)) & \text{if } c = 1 \end{cases} \tag{11}
$$

The model is trained by finding parameters $(\theta, w_0, w_1)$ that maximise the log-likelihood under the BTL model analogously to (10), replacing $r_\theta$ with $r_{\theta, w_0 w_1}$.

- **Adversarial:** The adversarial model adds an additional network $h_\phi : \mathbb{R}^{512} \to \mathbb{R}$ on top of the multihead architecture, mapping from $\hat{z}$ to unnormalised log probabilities of $C = 1$. Here, we let $h_\phi$ be a 1-layer MLP with a hidden dimension of 512. The model is trained under the adversarial objective:

$$
\min_{\theta, w_0, w_1} \max_\phi \mathcal{L}_\text{R}(\theta, w_0, w_1) - \lambda \mathcal{L}_\text{adv}(\theta, \phi), \tag{12}
$$

where $\mathcal{L}_\text{R}(\theta, w_0, w_1)$ is the negative log-likelihood under the BTL model of the reward function $r_{\theta, w_0, w_1}$ as in the Multihead model and $\mathcal{L}_\text{adv}$ is the binary-cross entropy loss computed across all prompt-response pairs–i.e., both for $(x, y)$ and $(x, y')$ within each sample $(x, y, y', c, \ell)$:

$$
\mathcal{L}_\text{adv}(\phi, \theta) = - \sum_{(x, y, y', c) \in \mathcal{D}} h_\phi(\hat{z}) c + (1 - h_\phi(\hat{z}))(1 - c) + h_\phi(\hat{z}') c + (1 - h_\phi(\hat{z}'))(1 - c), \tag{13}
$$

where $\hat{z} = g_\theta(x, y)$ and $\hat{z}' = g_\theta(x, y')$. The trade off between the two losses is controlled by the hyperparameter $\lambda$, which in our experiments we set to 1.0. The min-max optimisation problem is implemented with the gradient reversal technique (Ganin et al., 2016).

All MLPs are implemented with GELU activation functions (Hendrycks & Gimpel, 2023).

**Training.** All models are trained using the Adam optimiser with a learning rate of 1e-4 for 10 epochs. Model weights with the highest validation accuracy are saved for evaluation. For each value of $\rho$ we train models with 5 random seeds.

### E.2. Results

Table 2 shows the test-time accuracies of all models trained on datasets with varying values of $\rho$. Table 3 shows the accuracies on the training sets at the training step corresponding to the best model performance on the validation set. As discussed in the main body of the paper, the Base model exhibits strong overfitting. The Multihead architecture mitigates this to an extent, with the additional Adversarial objective bringing further improvements on the inconsistent samples.

| $\text{type}(X) = C$ | False | | | True | | |
|---|---|---|---|---|---|---|
| model | Base | Multihead | Adversarial | Base | Multihead | Adversarial |
| $\rho$ | | | | | | |
| 0.5 | $60.0 \pm 0.2$ | $62.9 \pm 0.2$ | $\mathbf{63.7} \pm 0.2$ | $60.9 \pm 0.1$ | $66.0 \pm 0.1$ | $\mathbf{66.3} \pm 0.1$ |
| 0.6 | $59.6 \pm 0.1$ | $62.4 \pm 0.1$ | $\mathbf{63.0} \pm 0.1$ | $61.3 \pm 0.1$ | $67.0 \pm 0.1$ | $\mathbf{67.3} \pm 0.1$ |
| 0.7 | $58.4 \pm 0.1$ | $60.6 \pm 0.1$ | $\mathbf{62.3} \pm 0.1$ | $62.5 \pm 0.2$ | $67.2 \pm 0.1$ | $\mathbf{67.6} \pm 0.1$ |
| 0.8 | $56.7 \pm 0.1$ | $58.6 \pm 0.1$ | $\mathbf{61.3} \pm 0.1$ | $63.7 \pm 0.2$ | $67.6 \pm 0.2$ | $\mathbf{67.9} \pm 0.2$ |
| 0.9 | $55.9 \pm 0.1$ | $56.3 \pm 0.1$ | $\mathbf{58.9} \pm 0.1$ | $64.2 \pm 0.2$ | $67.9 \pm 0.2$ | $\mathbf{68.4} \pm 0.2$ |
| 1.0 | $\mathbf{54.9} \pm 0.1$ | $53.8 \pm 0.1$ | $53.5 \pm 0.1$ | $64.1 \pm 0.2$ | $67.9 \pm 0.2$ | $\mathbf{68.4} \pm 0.2$ |

Table 2: Test accuracy [%] across all model architectures trained on datasets with varying values of $\rho$.

| $\text{type}(X) = C$ | False | | | True | | |
|---|---|---|---|---|---|---|
| model | Base | Multihead | Adversarial | Base | Multihead | Adversarial |
| $\rho$ | | | | | | |
| 0.5 | 73.5 ± 0.2 | 72.2 ± 0.2 | 72.2 ± 0.2 | 74.1 ± 0.2 | 74.8 ± 0.2 | 74.2 ± 0.2 |
| 0.6 | 71.3 ± 0.2 | 73.0 ± 0.2 | 73.0 ± 0.2 | 73.1 ± 0.1 | 75.8 ± 0.1 | 75.8 ± 0.1 |
| 0.7 | 73.9 ± 0.2 | 70.0 ± 0.2 | 68.0 ± 0.2 | 78.0 ± 0.1 | 74.9 ± 0.1 | 72.7 ± 0.1 |
| 0.8 | 69.4 ± 0.3 | 68.0 ± 0.3 | 69.9 ± 0.3 | 75.1 ± 0.1 | 75.3 ± 0.1 | 74.7 ± 0.1 |
| 0.9 | 73.7 ± 0.4 | 64.5 ± 0.4 | 67.2 ± 0.4 | 80.1 ± 0.1 | 75.0 ± 0.1 | 74.6 ± 0.1 |
| 1.0 | – | – | – | 79.6 ± 0.1 | 74.1 ± 0.1 | 72.4 ± 0.1 |

Table 3: Training accuracy [%] across all model architectures trained on datasets with varying values of $\rho$.

