# OpenReview forum: "Preference Learning for AI Alignment: a Causal Perspective"
_ICML.cc/2025/Conference — ICML 2025 poster_

### Official Review · Reviewer_Leyp · 2025-03-15

**Overall Recommendation:** 3

**Summary:**

This paper proposes a causal framework for preference learning in the context of aligning LLMs with human values. The authors argue that relying solely on observational data can lead to reward functions that pick up spurious correlations rather than true causal drivers of user preferences. To address this, the authors develop a model that incorporates causal inference principles such as potential outcomes, confounding, and latent treatments. Through theoretical analysis, they emphasize the critical assumptions needed for causal identification, most notably unconfoundedness and latent overlap, and discuss how these assumptions often fail in real-world data collection settings. Empirically, they demonstrate that strong correlations among latent features or user-specific objectives can cause overfitting and hinder the robustness of reward models under distribution shifts. The paper then proposes an “adversarial multi-objective reward model” to mitigate confounding effects, showing improved generalization, particularly in highly confounded scenarios.

## update after rebuttal
Thanks the authors for their detailed responses. My questions and concerns are fully addressed, and I will keep my original rating.

**Claims And Evidence:**

Yes. The paper provides detailed explanation on the problem formulation, theoretical statements, and numerical experiments.

**Essential References Not Discussed:**

No.

**Experimental Designs Or Analyses:**

The authors conducted case study over two public datasets. The analyses look fine to me, but the synthetic task design remain my key concern.

**Methods And Evaluation Criteria:**

The paper’s methods and evaluation align well with its theoretical goals with well-known dataset like HH-RLHF and UltraFeedback, but they mainly use synthetically augmented datasets instead of real-world data.

**Other Comments Or Suggestions:**

No.

**Other Strengths And Weaknesses:**

**Strengths**
A key strength is its clear explanation of the preference learning problem and seamless integration of a causal framework. It effectively connects user preferences, latent features, and rewards, making it easy to see how spurious correlations arise and hurt generalization. The solid theoretical backing on latent overlap and confounding, supported by a case study, reinforces these ideas. Overall, it offers a well-structured causal perspective.

**Weaknesses**
Some arguments in the paper seem either insufficiently supported or somewhat overstated. For example, the concern over "unobserved confounding" suggests that latent user attributes (like professional background) might create misleading correlations, such as academic-style queries leading to a preference for rigorous answers. However, in many real-world cases, these correlations are exactly what a model should capture. Medical experts, for instance, genuinely need detailed and precise explanations. If most academic queries come from users who actually require rigorous answers, then learning the pattern “academic → rigorous” isn’t necessarily a problem.

Regarding the “low overlap” issue, the paper argues that rare combinations of latent features can hurt generalization under distribution shifts and suggests extensive interventions to ensure broad coverage. However, in practice, dedicating significant resources to extremely rare cases may not always be worthwhile—focusing too much on edge cases could come at the cost of improving more common scenarios. The paper doesn’t address this trade-off, making its push for systematically controlling all latent factors feel somewhat idealistic in real-world data collection.

**Questions For Authors:**

See Weaknesses section.

**Relation To Broader Scientific Literature:**

This paper provides an interesting causal perspective over AI alignment, and discusses the key challenges critical assumptions for generalizing reward models to unseen texts and contexts.

**Theoretical Claims:**

The propositions in the main paper seems fine to me.

---

> ### Author Rebuttal · Authors · 2025-04-01
>
> Thank you for your review and for engaging with the core ideas of our work. We appreciate your time and feedback, and we’d like to address your concerns in hopes of clarifying our contributions and positioning.
>
> **The semi-synthetic design** We appreciate the reviewer’s attention to our experimental design and understand their concern about the use of semi-synthetic tasks. However, we’d like to clarify why semi-synthetic evaluations are a necessary and principled choice for studying causality in preference learning. Evaluation of causal inference methods demands knowledge of the true data-generating process (e.g., presence of confounders, the true set of causal variables), which is *unknowable* in purely observational real-world datasets. Semi-synthetic data allows us to:
>
> - Control the degree of overlap.
> - Inject controlled confounding (to simulate real-world biases).
> - Precisely measure how well the competing methods recover true relationships.
>
> Real-world datasets only show the observed human choices, not the counterfactual "what-if" responses needed to assess how a user would react if the same prompt were answered differently or if they acted according to a different objective. By augmenting these real-world public datasets with synthetic variations (e.g., simulating distribution shifts or conflicting objectives), we create a controlled testbed to isolate causal challenges—something impossible with static, observational data. Our approach aligns with established causal inference literature, where semi-synthetic data is the standard for evaluating methods [1, 2].
>
> **Useful vs. harmful correlations** We agree with your observation that correlations like “academic-style queries → rigorous answers" are often valid and desirable—indeed, in many cases, these patterns reflect genuine user needs.  However, our focus is on the challenges such correlations pose when training reward models from observational data where confounding is present. This issue is particularly critical in the context of steerable alignment, where we aim to gain control over which objective an LLM aligns to at inference time, rather than letting the model infer it purely from its input. Our case study in Section 4 examines the problem of learning a multi-objective, steerable reward model, where the two objectives considered are learned from different data distributions. We demonstrate that the naive approach fails when the train reward model is applied to data distributions different than at training time, which can plausibly occur in the real world. We also show how insights from causal representation learning can help build more robust reward models.
> **ACTION:** We acknowledge that not all correlations are harmful, and we will clarify this nuance in the revised version of our manuscript.
>
> **Practicality vs. coverage** You also raise a valid concern about the practicality of ensuring broad coverage of latent factors. We agree that exhaustive interventions are often infeasible, and we do not advocate for unrealistic data collection burdens. However, our goal is not to advocate for rigid control but rather to bring attention to challenges in learning across heterogeneous user populations, merging datasets collected with different objectives, or collecting feedback directly from users who wrote the prompts—practices that are present in preference learning today. While training reward models in an unsupervised, uncontrolled manner is flexible and scalable, we argue that the community should consider seeking a better balance between practicality and robustness. We highlight cost-effective strategies such as active querying (Appendix C) and causal regularisation techniques (as exemplified with the adversarial model) that can improve robustness without excessive overhead.
>
> **ACTION:** We will explicitly discuss the trade-offs between practicality and robustness in the final version, clarifying our position.
>
> ---
>
> We hope this response alleviates your concerns about our paper’s positioning. Our work does not dismiss observational data but instead provides tools to diagnose its limitations—a step we find crucial to achieving scalable and robust alignment. We believe the paper’s insights are valuable for the community, especially as alignment research moves towards greater personalisation.
>
> Thank you again for your time and feedback. We are happy to answer any further questions you may have and incorporate further revisions to ensure clarity in the final version.
>
> [1] arxiv.org/abs/1606.03976
>
> [2] arxiv.org/abs/1705.08821

---

### Official Review · Reviewer_tb1d · 2025-03-18

**Overall Recommendation:** 4

**Summary:**

- This paper introduces a causal framework for preference learning in AI alignment, specifically focusing on reward models trained on LLM prompts and response pairs. The authors frame prompt-response-response tuples as treatment variables, with latent rewards for each prompt-response combination serving as mediators for the observed binary preference label (the outcome variable).
- Reward models aim to learn to predict these latent rewards as a function of text, essentially estimating a treatment effect. However, these models may fail if they don't account for the underlying causal structure. The authors particularly focus on cases where contextual variables also affects the reward (and in some cases, the prompt too). For example, if a user's domain knowledge determines both the prompt and their assessment of responses, this creates confounding that precludes causal interpretation without additional assumptions. The paper draws on standard causal inference techniques and insights from causal representation learning to identify these necessary assumptions.

## update after rebuttal: the authors have addressed all of my concerns and suggestions.

**Claims And Evidence:**

- The paper's primary theoretical contribution is a causal framing of preference learning. The theoretical results about causal identification are standard results from the causal inference literature, adapted to the preference learning context and accounting for the latent structure of textual features. They claim to be the first to address confounding due to user-specific covariates, though they note previous work that has identified the existence of user-specific covariates and their effect on rewards [but not necessarily through confounding].

**Essential References Not Discussed:**

- In the discussion of causal representation learning on page 5 and in Appendix C, the authors should reference the fundamental challenges in learning causal representations from stationary data. Much theoretical work has been done demonstrating how identification of causal representations requires auxiliary labels (for instance, using paired examples where only a small number of ground-truth causal variables have been intervened upon). This is especially relevant to the paragraph “Active querying strategies and interventions” in Appendix C.
- See Challenging Common Assumptions in the Unsupervised Learning of Disentangled Representations by Locatello et al., https://arxiv.org/abs/1811.12359 and any more recent relevant work citing this paper.

**Experimental Designs Or Analyses:**

- The HH-RLHF study provides a very nice comparative analysis of reward models under confounded data.
- There could be more explanation of how the Multihead architecture incorporates the causal structure of the data. (The review suggests that the $\hat{z}$ being learned as a function of just $(x, y)$ means it doesn't use information from $c$, but it would be helpful to clarify that $c$ doesn't use information from $\hat{z}$ either, mirroring the conditional independence found in the underlying causal graph.)
- See my comments under “Other Strengths And Weaknesses,” where I suggest that the UltraFeedBack experiment may be unnecessary or out of place.
- The experiments don't address treatment effect heterogeneity, which is one of the paper's theoretical motivations.

**Methods And Evaluation Criteria:**

- The HH-RLHF study effectively addresses the motivating challenge of confounding in a semi-synthetic environment. The dataset is appropriate.
- See my comments under “Other Strengths And Weaknesses,” where I suggest that the UltraFeedBack experiment may be unnecessary or out of place.

**Other Comments Or Suggestions:**

- Page 2, line 86: “maximising” should be corrected to “minimizing.”
- Page 4, line 194: “underepresentation” should be corrected to “underrepresentation.”
- Page 5, line 260: ensure clarity regarding the distinction between causal discovery and causal representation learning.
- Though some motivation for the causal framing is presented in the introduction, some additional comments here may be helpful, perhaps by emphasizing the downstream role of reward models in alignment.

**Other Strengths And Weaknesses:**

- Strengths
    * The paper is very readable with clear visual cues and figures.
    * Extensive use of examples enhances understanding of the concepts.
- Weaknesses
    - Overlap/Positivity
        * Inconsistent terminology: The use of the term "overlap" in Assumption 3 and Assumption 5 to cover the case without conditioning on covariates is confusing, as this term is typically used to refer to overlap across covariate subpopulations (which the paper itself discusses on page 6, line 315). Even before this section, the paper describes the overlap assumption in the more standard "conditioning on covariates" way on page 4, line 202, noting that "The assumption of positivity requires that every user has a non-zero probability of being assigned any combination of texts given their covariates."
        * Unnecessary for identification: The overlap assumption as defined in Assumption 3 and Assumption 5 is not actually necessary for the proofs of Proposition 1 or Proposition 2. What the paper appears to be getting at is that overlap is not needed for causal identification in the case without confounding, but it is needed if one wants to estimate those quantities in a non-parametric form. If we assume a parametric model (e.g., a linear model), then we don't actually need to observe all possible combinations of X, Y, Y', contrary to what the paper claims.
        * Statistical vs. causal confusion: The UltraFeedBack Case Study (Appendix D, referenced in Table 1) is fundamentally about misgeneralization due to failure of the IID assumption. The experiments show how models fail when correlations present during training disappear at test time, but this doesn't necessarily involve causal misconceptions - it's about the IID assumption failing. A truly causal problem would exist if, even with IID data, infinite data would not allow us to estimate a treatment effect of interest. Including this study dilutes the paper's causal focus and may confuse readers about which problems are genuinely causal versus purely statistical.

**Questions For Authors:**

* Could you clarify why the UltraFeedback case study is framed as a causal problem rather than a statistical generalization issue? What makes the correlation shifts in this experiment fundamentally causal rather than just IID violations?
* Your theoretical motivation discusses both confounding and heterogeneous treatment effects, but your experiments focus primarily on confounding. Do you have plans to extend your empirical analysis to demonstrate the benefits of your causal approach for handling heterogeneous treatment effects?
* Your paper mentions that "confounding due to user-specific covariates has not been addressed in prior works." Could you elaborate on the distinction between your contribution and prior work that has identified the existence of user-specific covariates (like Siththaranjan et al.'s work on "hidden context")?

**Relation To Broader Scientific Literature:**

- The related works section nicely situates this paper within the broader causal inference and reward modeling literatures.

**Theoretical Claims:**

- Overall the theoretical claims are well-justified. But see my comments in “Other Strengths and Weaknesses.”

---

> ### Author Rebuttal · Authors · 2025-04-01
>
> Thank you for your detailed and insightful review. We appreciate the overall positive evaluation of our work. Below we address your comments and concerns:
>
> **The Multihead architecture** We appreciate your suggestion, **ACTION:** we clarify how the Multihead architecture incorporates the causal structure in the camera-ready version.
>
> **Additional References** Thank you for your suggestion regarding the additional references. **ACTION:** We suggest including [1, 2, 3, 4] as additional references strengthening our claims on page 5 and Appendix C.
>
> **Overlap/Positivity.**
>
> - **Terminology:** Preference learning conventionally assumes homogeneity of user preferences, so we stated identifiability assumptions in the *unconditional* case; this also reduces the notational clutter. However, we acknowledge that *overlap* is usually used conditionally. **ACTION:** We propose to rename Assumption 3 as *Unconditional Positivity* and Assumption 5 as *Unconditional Latent Positivity*. We will also clarify that we use *positivity* as shorthand for the unconditional case, while *Conditional Positivity (a.k.a. Overlap)* applies when user-specific covariates are considered (e.g., Example 4, Section 4).
> - **Sufficiency**:  Indeed, positivity is not necessary for identification but is required for non-parametric estimation as shown in the proofs of Proposition 1 and Proposition 2.  Thank you for pointing this out. **ACTION**: We will adapt the presentation of these results to make this clear.
> - **Statistical vs. causal confusion**: We agree with the reviewer that the UltraFeedback experiment primarily addresses statistical generalisation. The setup does not strictly violate the positivity condition, making the problem solvable in the limit of infinite data. Yet, a perfect correlation is not expected to occur in the real world so we focus on the practical challenge of *limited* latent positivity. It has been shown that near-violations of positivity inflate estimator variance and necessitate impractical amounts of data for robust inference [5, 6, 7]. Thus, while the observed performance degradation is a statistical issue, it stems from the weak satisfiability of the causal positivity assumption. **ACTION:** We appreciate the feedback and agree that the presentation of this experiment could be sharpened. We will explicitly frame this experiment as a statistical challenge arising from near-violations of the causal assumption and not a causal non-identifiability problem.
>
> **Other comments** Thank you for catching all the typos and your suggestions regarding the introduction. **ACTION:** we will implement them!
>
> **Heterogeneity of user preferences** We would like to highlight that while the case study of section 4 is presented as a study of confounding, the issues discussed are inherently a consequence of the heterogenous preferences of the two user groups: the helpfulness and the harmlessness preferring, where the two objectives are often *conflicting*.
> We find the investigation of more fine-grained variability in user preferences, going beyond the simplified setup of just two, an exciting direction for future work.
>
> **Distinction from prior work with user-specific covariates** Siththaranjan et al. highlight the possibility of heterogeneity in human preferences, pointing out the existence of potentially unobservable, hidden contexts that can influence user preferences. They demonstrate the negative consequences of implicitly aggregating over these hidden contexts when performing preference learning under a homogenous BTL model. Using our notation, they show that applying $\mathbb{E}[L(x; y, y')]$  as a global estimate of the expected preference for all users (instead of $\mathbb{E}[L(x; y, y') \mid C = c]$ separately for each set of user-specific covariates $c$) can lead to misalignment, especially with respect to the minority groups.
>
> However, their work does not address the issue of confounding, which arises when user-specific covariates $C$ not only influence the preference $L$ but also the distribution of the prompts $X$, making $C$ a confounder.  More generally, the limited number of related works considering user-specific covariates in preference learning assumes that user-specific preference datasets are collected under a randomised distribution of treatments. Our work challenges this assumption and explicitly considers the scenario where this distribution is influenced by $C$ (which is the case for instance when the prompts $X$ are written by the users themselves), leading to confounding issues examined in the case study of Section 4.
>
> ---
>
> We thank the reviewer for their positive evaluation of our work and are happy to answer any further questions!
>
> [1] arxiv.org/abs/1811.12359
>
> [2] arxiv.org/abs/2105.06422
>
> [3] arxiv.org/abs/2209.11924
>
> [4] arxiv.org/abs/2203.16437
>
> [5] doi.org/10.2307/2998560
>
> [6] doi.org/10.1111/1468-0262.00442
>
> [7] doi.org/10.3386/t0330

---

> > ### Comment · Reviewer_tb1d · 2025-04-04
> >
> > Thank you for your detailed reply! Your response addresses all of my comments and questions.

---

### Official Review · Reviewer_sdkh · 2025-03-18

**Overall Recommendation:** 4

**Summary:**

This paper uses a causal framework to articulate several assumptions commonly made when reward modeling from preference data. Namely, users are modeled as having implicit rewards which they assign to each response: hence, a preference label is formalized as being a function of these two potential rewards. The paper then investigates what assumptions are sufficient for the Bradley-Terry-Luce model to identify user preferences.
The first three assumptions are typical in causal inference (consistency, unconfoundedness, and positivity/overlap). Positivity is weakened to latent-overlap under an additional assumption of latent sufficiency. The paper suggests that these assumptions are not satisfied in practice by appealing to high-level, hypothetical examples (e.g. LLM responses aren’t expected to be simultaneously informal and professional).
There are two sets of experiments. Firstly, the practical relevance of the latent-overlap assumption is demonstrated by showing that linear reward models fail to generalize OOD on the UltraFeedback dataset. Secondly, the second suite of experiments demonstrates that confoundedness is by default an issue on the HH-RLHF dataset which can be addressed by careful architecture design when training the reward model (especially via a causality-inspired adversarial multi-objective reward model).

## update after rebuttal
The authors addressed my concerns to my satisfaction. I particularly appreciated the final rebuttal comment which elaborated on how a reward model could overfit conditionally to the confounder in "multi-objective" datasets. Hence, I am convinced that this really is saying something interesting about realistic training setups where many datasets are used during post-training: the key insight is that the way in which these datasets are mixed is itself providing information on which the reward model can overfit!

I am raising my score to 4, with the expectation that the authors will include these details in particular in the main body or appendix.

**Claims And Evidence:**

I worry that due to the presentation, many readers will misunderstand the significance of the assumptions in this paper as “necessary” when all that is shown is that they are “sufficient” for identification. Propositions 1 and 2 merely show that making these assumptions are sufficient, not necessary. The experiments show that under a particular training setup issues arise, but do not attempt to assess the scope of these issues “in the wild”: the experiments only demonstrate that misidentification can occur when these assumptions are violated on two datasets, the UltraFeedback dataset and HH-RLHF.
In particular, this paper should at least consider alternative assumptions which could provide identifiability guarantees while being more tenable for real-world preference learning setups. Specifically, consider the alternative assumption of “having access to observational data from multiple environments”, for which there is a rich literature demonstrating that there is rich signal (for both the causal identification task of Section 3.1, and causal representation learning task in “discovery of Z” in line 255 (right)). For instance, “Robust Agents Learn Causal World Models” https://openreview.net/pdf?id=pOoKI3ouv1, “Interventional Causal Representation Learning” https://arxiv.org/abs/2209.11924, Chapter 5 of “Identifiable Causal Representation Learning” https://arxiv.org/pdf/2406.13371, to name a few. This multi-environment assumption is more likely to be satisfied (thus providing identifiability) for leading reward models which train on multiple datasets, unlike the experiments here which only use singular datasets. This undermines the warnings for practitioners in e.g. Section 3.2.1 and Section 4 of this paper.

**Essential References Not Discussed:**

Line 173 (left): “To the best of our knowledge, confounding due to user-specific covariates has not been addressed in prior works.”

There is lots of work which view language models as inferring latent user attributes. e.g. “Language Models as Agent Models”: https://arxiv.org/pdf/2212.01681. Since in practice leading reward models are almost always fine-tuned LLMs, any work that discusses user-specific covariates in the context of LLMs is relevant for reward modeling.

**Experimental Designs Or Analyses:**

Please note how the experimental results in Section 4 do not localize confounding alone, as they also implicitly make a single-environment assumption. Do these results still hold even when models are trained on multiple datasets, as specified above?

**Methods And Evaluation Criteria:**

As discussed above, the experiments conducted here suffice for demonstrating that issues can arise, but are insufficient for claiming that these issues are *actually* observed for leading reward models and aligned LLMs. In practice, reward models are not trained on only a single dataset (as is the implicit assumption in the experiments).
Under the multi-environment assumption (that is, training a reward model across many different datasets simultaneously, as is done for leading reward models such as RLHFlow/ArmoRM-Llama3-8B-v0.1 https://arxiv.org/abs/2406.12845 which was trained on HelpSteer, UltraFeedback, BeaverTails-30k, CodeUltraFeedback, Prometheus, and various Argilla datasets), it seems quite possible that replicating the experiment in Section 4 will not yield the same negative results, even when inducing strong correlations in each dataset individually.

In summary: how much are the experimental results of Section 4 simply due to the restriction to single datasets?

**Other Comments Or Suggestions:**

Typo: line 20 (left) should be “causal” not “casual”.

**Other Strengths And Weaknesses:**

The paper is clearly organized, and can serve as a good introduction for practitioners not already familiar with causal representation learning.
The adversarial multi-objective reward model was neat.

Weaknesses have already been sufficiently noted elsewhere in this review.

**Questions For Authors:**

Do the results of Section 4 still hold under a multi-environment assumption, that is, even when models are trained simultaneously on multiple datasets, e.g. HelpSteer, UltraFeedback, BeaverTails-30k, CodeUltraFeedback, Prometheus, and various Argilla datasets as used for leading real-world reward models? Leading reward models are not trained on single datasets anymore, so it is not clear that the base model’s worse performance is due to confounding (as claimed) or of only using a single environment.

**Relation To Broader Scientific Literature:**

The causal assumptions and latent discussion are well-established approaches in the literature on causal representation learning.
I am not aware of any work which applies it to reward models in precisely the way done in this paper. There are somewhat close papers, namely 1. “RATE: Causal Explainability of Reward Models with Imperfect Counterfactuals” https://arxiv.org/abs/2410.11348v2, which introduces a causal framework for understanding and evaluating the spurious correlations of reward models regardless of whether they identify the original user preferences and leverages a similar latent variable framing and 2. “Aligning Large Language Models with Counterfactual DPO” https://arxiv.org/abs/2401.09566 which uses counterfactual pairs to address spurious correlations during the alignment process.
Given this context, the main contribution of this paper is in clearly articulating several sufficiency assumptions for identification of preferences. The assumptions themselves are unsurprising to anyone versed in causal inference, but may be new to reward model developers. (Which is precisely why it is important that this paper address that the assumptions here are sufficient, not necessary, and also highlight competing assumptions like multi-environment settings).

**Theoretical Claims:**

I checked the proofs of Propositions 1 and 2, which look fine: they use standard proof techniques from causal inference.

---

> ### Author Rebuttal · Authors · 2025-04-01
>
> Thank you for your thorough and insightful evaluation of our work. We appreciate the time and effort you put into assessing our paper. Below, we address your questions and comments point by point.
>
> **Presentation of assumptions** We acknowledge your concern regarding the potential misinterpretation of our assumptions as necessary rather than sufficient. **ACTION:** We will make appropriate adjustments in the phrasing in the revised version of our paper. In particular, we suggest that the first paragraph of section 3.1 is instead formulated as:
>
> > Causal inference provides a framework to answer counterfactual, 'what if' questions even when only observational data is available. A key part of causal analysis involves ensuring that the causal quantity of interest can be estimated from observational data alone. The following commonly made assumptions, adapted to our preference learning setup, are sufficient to guarantee identifiability and non-parametric estimability:
> >
>
> **The multi-environment setup** We greatly appreciate your suggestion to consider multi-environment setups and acknowledge their importance in broadening identifiability guarantees. **ACTION:** To further emphasise the fact that the assumptions discussed in our paper are sufficient rather than necessary, we will elaborate that alternative setups, such as having access to data from multiple environments, under certain conditions may also enable identifiability. We will reference the related literature you highlighted and contextualise it within multi-dataset reward model training. While learning reward models from multiple datasets is an emerging research area, its causal implications warrant further investigation. We view extending our framework to multi-environment settings as a promising avenue for future work.
>
> That said, our case study in Section 4 underscores a fundamental challenge in merging datasets collected under divergent objectives. Specifically, the HH-RLHF dataset comprises two independent subsets: the helpfulness subset and the harmlessness subset, which can be interpreted as distinct environments. Our experiments highlight potential pitfalls in training a multi-objective reward model when datasets exhibit distributional differences.
>
> We acknowledge the relevance of investigating our results in larger-scale multi-environment settings, such as those considered in ArmoRM. However, the datasets involved in ArmoRM lack counterfactual preference labels—i.e., they provide preferences only under their respective training objectives but do not indicate how preferences would transfer across datasets. The lack of overlap in objectives between the datasets of ArmoRM makes it difficult to assess generalisation beyond the dataset-specific correlations present at training time. In our experiments, the assessment of the counterfactual scenarios is possible thanks to the provision of additional, counterfactual labels to the HH-RLHF by [1]. The presented results exemplify potential risks associated with training multi-objective, steerable reward models from a composition of datasets collected under different, potentially conflicting objectives. We hope this motivates the development of new datasets tailored for robust reward model training.
>
> **Related work & contributions** We appreciate your pointers to related work. **ACTION:** we will incorporate the suggested references into our discussion of related literature. Additionally, we would like to point out that our contribution extends beyond articulating sufficiency assumptions for preference identification. A second key contribution of this work is the highlighting of the potential confounding effects due to user-specific covariates.
> While there indeed exist prior works that aim to infer user-specific covariates (e.g., [2, 3]), they do not examine the confounding issue.  The implicit assumption of prior works is that the covariates $C$ only influence the preference label $L$ but not the prompt distribution $X$. Our work considers cases where $C$ influences both $X$ and $L$, and thus it acts as a confounder. This is the case when users write the prompts $X$ and subsequently score the LLM’s generated responses to these prompts, which may in practice be the case (e.g. the HH-RLHF dataset).
>
> **Minor corrections** Thank you for catching the typo on line 20, we will correct it!
>
> ---
>
> We are grateful for the opportunity to refine our work based on your thoughtful feedback. Your suggestions have helped us improve the presentation clarity, depth and broader impact of our research. We hope our explanations and proposed revisions address your concerns. Thank you for your time and consideration.
>
> [1] arxiv.org/abs/2312.08358
>
> [2] arxiv.org/abs/2402.05133
>
> [3] arxiv.org/abs/2409.11901

---

> > ### Comment · Reviewer_sdkh · 2025-04-04
> >
> > > While learning reward models from multiple datasets is an emerging research area, its causal implications warrant further investigation. We view extending our framework to multi-environment settings as a promising avenue for future work.
> > That said, our case study in Section 4 underscores a fundamental challenge in merging datasets collected under divergent objectives. Specifically, the HH-RLHF dataset comprises two independent subsets: the helpfulness subset and the harmlessness subset, which can be interpreted as distinct environments. Our experiments highlight potential pitfalls in training a multi-objective reward model when datasets exhibit distributional differences.
> >
> > Question: Can you elaborate on how your case study in Section 4 highlights potential pitfalls when datasets exhibit distributional differences, as you say HH-RLHF exhibits?
> >
> > Note: I acknowledge that when RLHF was first introduced (https://arxiv.org/abs/1706.03741), reward models were trained on homogeneous data, but in practice all of the leading models on RewardBench have been fine-tuned using multi-objective data. So the relevance gap is very significant, and the paper would be much stronger if it addressed this. However, I acknowledge that this lies outside the current scope of the paper.

---

> > > ### Author Response · Authors · 2025-04-04
> > >
> > > Dear Reviewer,
> > >
> > > Thank you for your question. Due to character constraints in the main rebuttal, we could only briefly reference this issue. We’re happy to elaborate below:
> > >
> > > As described in [4], the HH-RLHF dataset was constructed by combining *two distinct datasets* collected under different objectives: *helpfulness* and *harmlessness*. As observed by [1], this resulted in the two datasets having distinct distributions of prompts:
> > >
> > > > *"… we found that the distribution of prompts was quite different between the helpfulness and harmlessness splits of the dataset. In the helpfulness split, most prompts were harmless questions or requests for assistance. In contrast, in the harmlessness split, most prompts were specifically chosen to elicit harmful behavior."*
> > > >
> > >
> > > This data collection setup induced a correlation between the labelling objective $C$ (helpful vs. harmless) and the prompt type, which we denote by $\mathrm{type}(X)$. As a result, the objective $C$ acts as a *confounder*—any learned reward model could exploit the correlation between $C$ and $\mathrm{type}(X)$ to predict the objective-conditioned preferences: $\mathbb{E}[L(x; y, y') \vert C=c]$.
> > >
> > > To study this effect, we use the synthetic counterfactual labels from [1], which allow each sample $(x, y, y')$ to be evaluated under both objectives: the “factual” ones where the prompt type agrees with the objective (as in the original dataset), and the “counterfactual” ones where $\mathrm{type}(X) \neq C$. By controlling the training time correlation $\rho := P(\mathrm{type}(X) = C)$, our experimental setup lets us isolate how well the reward models generalise beyond their training distribution.
> > >
> > > Importantly, at $\rho = 1.0$—corresponding to the original HH-RLHF dataset—the reward models tend to overfit: they perform well when evaluated on instances and objectives for which $\mathrm{type}(X) = C$, but fail on the *counterfactual* ones where $\mathrm{type}(X) \neq C.$ As $\rho$ decreases, this effect is less severe. At $\rho=0.5,$ i.e. when the distribution of prompts for learning each objective is perfectly overlapping, all reward models perform well on both types of evaluations.
> > >
> > > In practice, however, when learning from static, observational datasets, $\rho$ is not controllable—posing risks for the robustness of learned reward models to test-time distribution shifts. Our results highlight a potential challenge in multi-objective, multi-environment preference learning: if each objective is learned from a distinct distribution of prompts (effectively, different environment), reward models may overfit to spurious environment-specific features rather than correctly recognising the true underlying factors driving objective-conditioned preferences.
> > >
> > > Our case study motivates, for example, the use of targeted interventions during data collection to increase overlap between prompt distributions across different objectives or the introduction of appropriate regularisers—such as in the Adversarial model presented in this study.
> > >
> > > We hope this response clarifies our reasoning. Thank you again for engaging with our work.
> > >
> > > [1] *Distributional Preference Learning: Understanding and Accounting for Hidden Context in RLHF* – https://arxiv.org/abs/2312.08358
> > >
> > > [4] *Training a Helpful and Harmless Assistant with Reinforcement Learning from Human Feedback* – https://arxiv.org/pdf/2204.05862

---

### Official Review · Reviewer_1738 · 2025-03-21

**Overall Recommendation:** 3

**Summary:**

This paper introduces a causal framework for preference learning in AI alignment, aiming to improve the robustness of reward models. Reward modeling from preference data is a crucial step in aligning large language models (LLMs) with human values. The authors propose integrating causality into reward modeling, arguing that this approach enhances model robustness.

A key challenge in preference learning is that pairwise preference datasets are often collected opportunistically. LLM users both evaluate model responses and generate prompts, introducing individual-specific confounders that bias the learning process. The paper illustrates this issue with examples and experimental results, demonstrating the vulnerability of naïve reward models to OOD (out of distribution) samples.

To address these challenges, the authors propose controlled, randomized experiments, where prompt-response pairs are allocated randomly across a representative population. Additionally, they suggest that preference data collection methods can be improved to infer user-specific objectives through: Explicit feedback: Users provide rationales for their preferences, offering richer insights into their underlying objectives and Contextual information modeling: Incorporating external context to refine preference interpretation. This causal perspective provides a more systematic and reliable approach to reward modeling, ultimately strengthening AI alignment with human values.

**Claims And Evidence:**

Please refer to the Questions For Authors Section.

**Essential References Not Discussed:**

Please refer to the Questions For Authors Section.

**Experimental Designs Or Analyses:**

Please refer to the Questions For Authors Section.

**Methods And Evaluation Criteria:**

Please refer to the Questions For Authors Section.

**Other Comments Or Suggestions:**

Please refer to the Questions For Authors Section.

**Other Strengths And Weaknesses:**

Strengths:
Please refers to the Relation To Broader Scientific Literature section.

Weaknesses:
Please refer to the Questions For Authors Section.

**Questions For Authors:**

1. I did not fully understand the counterfactual outcome associated with not receiving the (X, Y, Y') treatment. Could you provide an intuitive example? For instance, in the case of drug treatment effects, the counterfactual outcome represents the difference in health status of the same individual under treatment versus without treatment. Typically, we compare the observed outcome under treatment with the hypothetical (counterfactual) outcome the individual would have had if they had not received the drug. However, in this paper, it seems that the author defines the expected outcome L(X,Y,Y′) as the counterfactual outcome. This approach is not intuitive to me—could you clarify the reasoning behind this definition?
2. Lack of Supporting Evidence
In Line 14, the authors state:
"Moreover, pairwise preference datasets are often collected opportunistically, with LLM users both evaluating model responses and generating the prompts eliciting them."

Could you provide references to studies that actually collect data in this manner? This is a key claim, as the authors argue that current practices in preference data collection are problematic. However, without supporting evidence, it is difficult to assess the validity of this concern. I would appreciate citations or empirical examples to substantiate this argument.

Questions on examples and case studies

1. Example 1 shows the existence of confounding X. But why not model it as contextual dueling bandits? Such as the modeling of Nearly optimal algorithms for contextual dueling bandits from adversarial feedback?
2. Example 2, the latent effect model, is fine. However, I am uncertain about how this approach will balance model misspecification and robust generalization. This modeling framework seems to assume precise knowledge of the reward model, which raises the question: does this approach truly enhance robustness? Could the authors clarify how the proposed modeling method mitigates misspecification while still ensuring robust generalization? Since in real world, model misspecification will be very common and it is possible that overfitting performs even better than identifying "wrong" or "incomplete" casual relationship.
3. Table 1 highlights the issue, but I find the result unsurprising—OOD (out-of-distribution) performance degradation is expected. However, is there any evidence that modeling reward causally can actually alleviate this problem? Additionally, does similar behavior occur in other datasets, or is this effect specific to the dataset used? Could you clarify why UltraFeedback was chosen for evaluation? A broader justification or comparison with other datasets would strengthen the argument.
4. Case Study Results. I believe it is crucial to include experimental results for Direct Preference Optimization (DPO) as well. My main concern is that reward modeling is not necessarily required for effective preference learning. If DPO is used instead of reward modeling, would the same concerns about robustness and generalization still apply? Including a comparison with DPO would provide a stronger evaluation and help clarify whether reward modeling is essential in this context.

I am willing to reevaluate my rating if all my questions are clearly addressed.

**Relation To Broader Scientific Literature:**

It opens new research directions, sparks discussions on integrating causal frameworks into reward modeling to enhance robustness, and provides valuable guidance on observational data collection processes. I believe this is a meaningful addition to the broader literature.

**Theoretical Claims:**

I did not find any apparent errors in the author's mathematical derivations. However, the theoretical content in this paper is relatively light. I reviewed the derivations of Proposition 1 and Proposition 2, and they appear to be correct. That said, I recommend that the authors provide better citations to relevant literature and more discussion on key intuitions, especially for readers unfamiliar with causal representation learning.

For instance, in Proposition 1, it would be helpful to clarify why the expectation in Line 149 suggests that observed statistical associations have a causal interpretation. Providing additional context or references could strengthen the argument and make the results more accessible to a broader audience.

---

> ### Author Rebuttal · Authors · 2025-04-01
>
> Thank you for the time taken to evaluate our work. We your recognition of our work as a meaningful addition to the broader literature. Below we address your comments:
>
> **Presentation.** In the camera-ready version, we will use the additional space to clarify key intuitions, especially for readers less familiar with causal representation learning, and include citations to introductory causality resources. Thank you for this suggestion.
>
> **Q1: Treatment effects.** Thank you for your question regarding our causal framework. Due to space constraints, we have provided an intuitive explanation [here](https://imgur.com/a/YqzO5KG).
>
> **Q2: Evidence.** Thank you for prompting us to substantiate the claim about data collection practices. We are happy to provide supporting evidence:
>
> 1) the widely-used Anthropic HH-RLHF dataset provides concrete evidence of this. As described in [2], the dataset construction process explicitly involved users both writing prompts and providing preference judgments on LLM’s responses.
>
> 2) OpenAI's ChatGPT interface empirically demonstrates this collection approach. Users are frequently presented with pairs of responses to their own queries and asked to indicate preferences. OpenAI's privacy policy confirms these interactions are used for model improvement.
>
> **Q1: Bandits.**  Thank you for the suggestion to consider contextual dueling bandits. However, we clarify that the confounder in Example 1 is the unobserved user-specific variable $C$, which influences both the prompt $X$(during dataset generation) and the preference labels **$L$**—not $X$ itself. Secondly, contextual dueling bandits focus on optimising a selection policy for choosing responses **$Y,Y'$** given the observable context $X$ (the prompt), whereas in our setup, these responses are passively sampled from an LLM, whose policy we do not control. Our work focuses on learning a reward model from static, observational data. While applying a contextual dueling bandit framework to jointly learn preferences and control the sampling policy is an interesting direction (e.g. [3]), our work focuses on biases in preference learning from observational data rather than on designing a strategy for sampling the candidate responses. Furthermore, even in a dueling bandits, preferences are typically modelled with the BTL model, which as discussed, does not account for confounding. If an unobserved $C$ influences both the prompts $X$ and preference labels $L$, the learned preference model would remain biased.
>
> **Q2: Example 2.** We would like to clarify that Ex. 2 is not intended to prescribe a fixed functional form for the reward model for learning. It is an illustration showing why conditioning on prompt-specific features ($Z^X$) is essential. The provided equation exemplifies a possible structure of the ground-truth reward function to highlight the plausibility of the effects of $Z^X$ being non-additive.
>
> In practice, to learn a reward model we would not assume knowledge of the parametric form or the set of relevant latent features. Instead, as discussed in the paragraph *Discovery of Z*, the latent factors are learned from data. This flexibility, however, induces challenges associated with causal discovery under minimal supervision [4, 1], including overfitting to spurious correlations, as illustrated by our case studies.
>
> **Q3: The UltraFeedback experiment.** This experiment highlights the impact of the latent overlap assumption—when strictly violated, causal identification is impossible. We show that weaker overlap makes the recovery of the ground-truth model significantly more challenging. This experiment highlights the inherent limitations of learning from observational data only, rather than promoting a specific approach.
>
> UltraFeedback was chosen for its unique structure, allowing control over correlations between latent features. Unlike other datasets like Stanford Human Preferences or HH-RLHF, which label instances based on a single objective, UltraFeedback scores each prompt-response pair across four latent factors.
>
> **Q4: DPO.** Thank you for this suggestion. While DPO bypasses the explicit reward modelling of RLHF, it defines an *implicit reward function* by shaping the policy’s logits to match preference probabilities according to the same BTL model as used in RLHF. Consequently, we expect DPO to inherit the same generalisation failures in OOD settings. While an empirical DPO comparison could offer additional insights, we prioritised analysing reward modelling, as RHLF remains the predominant approach in alignment and DPO is based on the same underlying assumptions.
>
> ---
>
> Thank you for your valuable feedback, which helped us improve the paper.  We hope our answer resolves your concerns, and we’d be grateful if you could reconsider your initial rating to help us make a meaningful addition to the literature.
>
> [1] arxiv.org/abs/2102.11107
>
> [2] arxiv.org/abs/2204.05862
>
> [3] arxiv.org/abs/2402.00396
>
> [4] arxiv.org/abs/1811.12359

---

> > ### Comment · Reviewer_1738 · 2025-04-03
> >
> > Thank you for the author's detailed responses. Most of the reviewer's concerns have been addressed. However, the reviewer remains particularly concerned about the absence of DPO experimental results as a baseline. Including DPO would significantly strengthen the empirical evaluation, as it would help demonstrate the importance of reward modeling and the robustness of the proposed approach. The reviewer would increase his rating if such results were included.

---

> > > ### Author Response · Authors · 2025-04-09
> > >
> > > Dear Reviewer,
> > >
> > > Thank you again for your feedback. We are happy to hear our response addressed your concerns. Following your suggestion, we attempted to include a DPO baseline. However, despite our best efforts, we encountered practical challenges that prevented us from obtaining meaningful results:
> > >
> > > **Computational constraints.** DPO requires LLM policy fine-tuning, which is computationally intensive. Given the compute available at our institution, we were only able to perform LoRA fine-tuning with small batch sizes (8–16), which are significantly smaller than those we could use for training the reward models based on the pre-computed LLM embeddings (i.e., bs=128).
> > >
> > > **Policy conditioning.** In our setup (Section 4), conditioning on the user objective $C$ is crucial. In reward modelling, this is achieved by passing $C$ at the input level—either by concatenating a one-hot encoded objective with the prompt-response embedding (Base model), or by directly incorporating $C$ into the model architecture (Multihead model). Enabling a similar form of conditioning in DPO is non-trivial and constitutes a significant challenge in itself. It can be, for instance, attempted at the input level ([5]) or within the parameter space of the LLM (e.g., via so-called steering vectors [6]). We tried adopting an input-level prompt-based approach (similar to [5]) to encode the objective:
> > >  ```
> > > Objective: <helpfulness / harmlessness>
> > > User: <prompt>
> > > Assistant: <response>
> > > ```
> > > Nevertheless, in our setting, this method did not lead to successful convergence of DPO training under the available resources. This suggests that prompt-based conditioning may be suboptimal for multi-objective policy learning—especially in settings where the objectives are in many instances conflicting. It also highlights a broader point: identifying a suitable and practical conditioning method for conditional policy optimisation is a separate research question, which we consider beyond the scope of this work.
> > >
> > > While we would have liked to include DPO as a baseline, we were unable to obtain conclusive results. Nonetheless, we note that the practical challenges we encountered—particularly around conditioning the LLM policy—are orthogonal to the core focus of our work: analysing the BTL model and preference learning practices from a causal perspective and identifying their potential pitfalls, such as the confounding effects exemplified in the Case Study. Our core theoretical arguments regarding preference learning remain conceptually applicable to any framework based on the BTL model.
> > >
> > > We hope this clarifies our rationale for the empirical focus on RLHF in this work, and we believe the insights and experimental evidence provided remain valuable and informative.
> > >
> > > Thank you for taking the time to engage with our submission, we really appreciate your feedback.
> > >
> > > Kind regards,
> > >
> > > The Authors
> > >
> > > ---
> > >
> > > [5] *Aligning to Thousands of Preferences via System Message Generalization* (NeurIPS 2024)
> > >
> > > [6] *Personalized Steering of Large Language Models: Versatile Steering Vectors Through Bi-directional Preference Optimization* (NeurIPS 2024)

---

### Decision · Program_Chairs · 2025-05-01

**Decision:**

Accept (poster)

**Comment:**

This paper applies causal thinking to the learning of reward models. Following an extensive discussion, all reviewers agree it constitutes an interesting and possibly impactful perspective on understanding reward models. The authors should incorporate the reviewer comments into the final version (per the discussion).